



# Improvement of inorganic aerosol component in PM₂.₅ by constraining aqueous-phase formation of sulfate in cloud with satellite retrievals: WRF-Chem simulations

Tong Sha[1], Xiaoyan Ma[1*], Jun Wang[2], Rong Tian[1], Jianqi Zhao[1], Fang Cao[3], Yan-Lin Zhang[3]

[1] Collaborative Innovation Center on Forecast and Evaluation of Meteorological Disasters (CIC-FEMD)/Key Laboratory for Aerosol-Cloud-Precipitation of China Meteorological Administration, Nanjing University of Information Science & Technology, Nanjing 210044, China

[2] Department of Chemical and Biochemical Engineering/Center for Global and Regional Environmental Research, The University of Iowa, Iowa City, IA, USA

[3] Yale–NUIST Center on Atmospheric Environment, International Joint Laboratory on Climate and Environment Change (ILCEC)/Jiangsu Provincial Key Laboratory of Agricultural Meteorology, College of Applied Meteorology, Nanjing University of Information Science & Technology, Nanjing 210044, China



**Abstract**
High concentrations of $PM_{2.5}$ in China have caused severe visibility degradation
and health problem. However, it is still a big challenge to accurately predict $PM_{2.5}$ and
its chemical components in the numerical model. In this study, we compared the
inorganic aerosol components of $PM_{2.5}$ (sulfate, nitrate, and ammonium (SNA))
simulated by WRF-Chem with in-situ data during a heavy haze-fog event (November
2018) in Nanjing. The comparisons show that the model underestimates the sulfate
concentrations by 81 % and fails to reproduce the significant increase of sulfate
concentrations from early morning to noon, which corresponds to the timing of fog
dissipation, suggesting that the model underestimates the aqueous-phase formation of
sulfate in clouds. In addition, the model overestimates both nitrate and ammonium
concentrations by 184 % and 57 %, respectively. These ultimately result in the
simulated SNA 77.2 % higher than the observations. However, as the important
aqueous-phase reactors, cloud water are simultaneously underestimated by the model.
Therefore, the modeled cloud water was constrained based on the MODIS Liquid Water
Path (LWP) observations. Results show that the simulation with MODIS-corrected
cloud water amount increases the sulfate by a factor of 3, decreases NMB by 53.5 %,
and can reproduce its diurnal cycles, i.e. the peak concentration at noon. Also, the model
absolute bias of nitrate decreases from 184 % to 50 %, especially for the nocturnal
concentrations, which suggests the MODIS-constrained simulation improved the
diurnal pattern. Although the simulated ammonium is still higher than the observation,
corrected cloud water lead to the decrease of the modeled bias of SNA from 77.2 % to



14.1 %. The strong sensitivity of simulated SNA concentration to the cloud water
provides an explanation for the bias of SNA simulation. Hence, the uncertainties of
cloud water can lead to model bias in simulating SNA, and can be reduced by
constraining the model with satellite observations.



## 1 Introduction

Severe and persistent haze pollution with daily concentrations of $PM_{2.5}$ exceeding
the Chinese standard of 75 μg m$^{-3}$, occurs frequently in China during recent decades,
which has aroused wide public attention due to its adverse impact on air quality,
regional and global climate, and human health (Huang et al., 2014). According to
previous studies, stagnant meteorological conditions with high atmospheric relative
humidity and low boundary layer height, high emissions of primary air pollutants, as
well as the rapid formation of secondary inorganic aerosols, including sulfate, nitrate,
and ammonium (SNA), are considered to be the main factors leading to the haze
episodes (Liu et al., 2020a). Earlier studies showed that the contribution of SNA to total
$PM_{2.5}$ mass concentration was over 50% during the severe haze events (Cheng et al.,
2016; Xu et al., 2017; Wang et al., 2019).
The chemical transport models (CTMs) are often used to predict the $PM_{2.5}$ pollution
and evaluate the emission control strategies. Most models show reasonable
performance on simulating surface $PM_{2.5}$ concentrations in China but perform poorly
on simulating the proportion of chemical components in $PM_{2.5}$, especially during the
severe haze periods (Gao et al., 2018; Chen et al., 2019). Many recent studies have
reached an agreement that CTMs generally underestimate sulfate concentrations but
overestimate nitrate concentrations (Wang et al., 2013; Wang et al., 2014; Zheng et al.,
2015a; Chen et al., 2016; Cheng et al., 2016; Fu et al., 2016; Gao et al., 2016; Li et al.,
2018a; Chen et al., 2019; Sha et al., 2019). The uncertainties such as meteorological
fields (Bei et al., 2017; Li et al., 2017c; Su et al., 2018), emission inventories (Ma et al.,



2018; Zhang et al., 2018; Qu et al., 2019), and parameterizations of physical and
chemical processes in the model (Gao et al., 2018; Luo et al., 2019; Alexander et al.,
2020), can contribute to the discrepancies of SNA and $PM_{2.5}$ between the models and
observations.

The underestimation of sulfate in the models has been mainly attributed to the

incomplete and/or inaccurate chemical mechanism. Generally, sulfate is formed
through the gas-phase oxidation of $SO_2$ by OH radicals, and aqueous-phase oxidation
of S(IV) (= $SO_2 \cdot H_2O + HSO_3^- + SO_3^{2-}$) by various oxidants (e.g., $H_2O_2$, $O_3$, $NO_2$, and $O_2$
(transition-metal-ion (TMI) catalysis)) in cloud droplets and aerosol water (the latter
often called the heterogenous reaction) (Cheng et al., 2016; Liu et al., 2020a). It is worth
noting that high atmospheric RH facilitates sulfate formation and aggravates the haze
pollution (Xue et al., 2016; Tie et al., 2017; Wu et al., 2019). Therefore, the formation
of sulfate is mainly through gas-phase reactions under relatively low atmospheric RH
(RH < 30 %), but through heterogeneous and aqueous-phase reactions under relatively
high atmospheric RH (RH > 60 %) (Li et al., 2017a). However, the mechanisms of
sulfate formation at high RH is still controversial and unclear (Cheng et al., 2016; Wang
et al., 2016; Ge et al., 2017; Guo et al., 2017; Liu et al., 2017; Yang et al., 2017; Li et
al., 2018b). Previous studies proposed that the oxidation of $SO_2$ by $NO_2$ in aerosol water
with almost neutral aerosol pH values (5.4-7.0) plays a dominant role in sulfate
formation during the severe haze episodes (Cheng et al., 2016; Wang et al., 2016).
However, the aerosol pH calculated by the ISORROPIA II model was moderately acidic
with the value of 3.0-4.9, suggesting that the pathway of $SO_2$ oxidation by dissolved



NO₂ was not important during the haze events in China (Guo et al., 2017; Ding et al.,
2019). Latest studies suggested that SO₂ heterogeneous reaction via TMI-catalyzed
oxidation perhaps dominates the sulfate formation during the haze periods, which is
also verified by the observations of sulfate oxygen isotopes (Shao et al., 2019). Since
the observations of the concentration, complexation, and solubility of TMI are not
available, the mechanism still remains unclear (Jacob, 2000; Wang et al., 2020). In order
to tackle the underestimation of sulfate in the model during the haze events, most
studies add the SO₂ heterogeneous reaction in the model, which is usually
parameterized as a reactive uptake process and assumed to be irreversible (Wang et al.,
2014; Zheng et al., 2015b; Chen et al., 2016; Li et al., 2017a; Feng et al., 2018; Li et
al., 2018a; Sha et al., 2019; Shao et al., 2019). Although the implementation of SO₂
heterogeneous reactions in the model can achieve an agreement of simulated and
observed sulfate concentrations during the haze episodes, the model still underestimates
the sulfate due to uncertainties of the parameters in this reaction, such as the pH, water
content and surface area of aerosol, as well as the gas uptake coefficients on aerosol
water.

Cloud/fog droplets can act as efficient reactors in which dissolved SO₂ reacts with

oxidations to form sulfate. Many studies showed that sulfate concentrations would be
enhanced by the occurrence of cloud and fog compared to the cloud-free conditions
(Crahan et al., 2004; Sorooshian et al., 2006; 2007; Wonaschuetz et al., 2012; Ervens
et al., 2018a). Previous modeling studies concluded that a major fraction of sulfate (60-
90%) is formed via aqueous (in-cloud) chemistry globally (Barth et al., 2000; Ma and





Salzen, 2006; Harris et al., 2013; Kim et al., 2015; Ervens et al., 2018b; Dovrou et al.,
2019). The aqueous formation rate depends on liquid water content (LWC), the size
distribution, pH and lifetime of cloud droplets, as well as the availability of oxidants.
The kinetic and mechanistic parameters that characterize the uptake processes of sulfate
precursors and oxidants, as well as the chemical reactions leading to sulfate formation
in the aqueous phase, are relatively well constrained in the model, therefore the largest
uncertainties in predicting in-cloud sulfate formation do not originate from the
understanding of the chemical processes, but from the prediction of cloud
microphysical and dynamical parameters, such as LWC and cloud lifetime (Rasch et al.,
2000; Ervens et al., 2015). Mueller et al. (2006) found that the simulated sulfate
concentration significantly increased after correcting the underestimation of model
cloud fraction. Xie et al. (2019) showed that the improvement in cloud fields in
MERRA-2 can eliminate approximately half of the bias in the surface sulfate
concentration during summertime relative to the MERRA data. However, only a few
studies focus on the sulfate underestimation caused by the bias of cloud fields during
the haze episodes. Therefore, a better understanding of the sensitivity of sulfate
simulations to cloud water is needed to improve the model performance on predicting
$PM_{2.5}$.

A persistent high $PM_{2.5}$ level accompanying the fog event (short as haze-fog event)

occurred in the Yangtze River Delta from 26 October to 2 December 2018. We choose
this period to investigate the impact of cloud/fog water on simulating SNA using the
WRF-Chem Model. The paper is organized as below. Section 2 shows the descriptions



of the model and data, as well as the meteorology evaluation. The evaluation of
simulated chemical fields and cloud water with observations, and sensitivity
experiments to study the impact of corrected cloud water on simulated SNA are
presented in section 3 and 4. Section 5 shows the summaries.
**2    Model configurations, data description, and model evaluation**
**2.1 Model configurations**
The WRF-Chem version 3.9.1 (Grell et al., 2005) is used in this study to conduct
the simulations on a domain over the eastern China with the horizontal resolution of 27
km and nested to a domain with 9 km covering the YRD (Fig. 1(a)). There are 42
vertical levels, with 24 levels below the boundary layer (about 1500 m) and the lowest
level about 21 m. The physical parameterization schemes include Lin microphysical
scheme (Chen and Sun, 2002), Grell 3-D cumulus scheme (Grell and Dezső, 2002),
RRTM (Mlawer et al., 1997) for longwave radiation and Goddard scheme for shortwave
radiation (Chou and Suarez, 1994), Yonsei University planetary boundary layer
parameterization (Hong et al., 2006), QNSE surface layer scheme (Sukoriansky et al.,
2005) and Noah land surface model (Tewari et al., 2004).
The Carbon Bond Mechanism (CBMZ) for gas-phase chemistry (Zaveri and Peters,
1999) and Model for Simulating Aerosol Interactions and Chemistry (MOSAIC)
aerosol module with 4 sectional aerosol bins and aqueous reactions (Zaveri et al., 2008)
are chosen in our study. MOSAIC predicts all the major aerosol species, including
sulfate, nitrate, ammonium, BC, primary organic mass, chloride, sodium, other
inorganic mass (OIN), and liquid water. Detailed descriptions of the SNA formation





mechanisms in the standard model can be found in Sha et al. (2019).
The 0.25°×0.25° National Center for Environmental Prediction's (NCEP) Final
Analysis (FNL) dataset (http://rda.ucar.edu/datasets/ds083.2/) provides the
meteorological initial and boundary condition. Anthropogenic emissions are taken from
Multi-resolution Emission Inventory for China (MEIC: http://www.meicmodel.org/)
for the year 2016 (Li et al., 2017b). The simulation starts on 24 November and ends on
2 December 2018, with the first 48 hours as the spin-up period.
**2.2 Observational data**
Meteorological variables are measured every three hours from five weather stations
in Nanjing, and obtained for this study from the Meteorological Information
Comprehensive Analysis and Process System (MICAPS) (green triangles in Fig. 1(b)),
which are used to evaluate the model performance on simulating meteorological fields.
The data include air temperature and relative humidity at 2m (T2, RH), wind speed and
direction at 10m (WS10, WD10), visibility (VIS), and accumulated precipitation (PRE)
(only the sample frequency of precipitation is 6 hourly). For surface pollution, two data
sets are used: (1) the hourly $SO_2$, $NH_3$, $HNO_3$, HONO, and inorganic chemical
components in $PM_{2.5}$ (sulfate, nitrate, and ammonium) concentrations measured by the
In-situ Gas and Aerosol Compositions monitor (IGAC) (Young et al., 2016) at Nanjing
University of Information Science & Technology (NUSIT) (32.2º N, 118.7º E; 22m
above sea level) (the blue circle in Fig. 1(b)); (2) the routine measurements of hourly
$NO_2$ and $PM_{2.5}$ concentrations at Maigaoqiao monitoring site (32.1º N, 118.8º E) in
Nanjing from the China National Environmental Monitoring Center (CNEMC) (since



the NUIST site did not observe $NO_2$ and $PM_{2.5}$ simultaneously, the observation data
from Maigaoqiao site nearest to the NUIST were used, shown as the red circle in Fig.
1(b)). Himawari 8 satellite data are used to represent the spatial area of this fog event
(https://www.eorc.jaxa.jp/ptree/index.html). Fog area is mainly indicated by the albedo
at three visible bands: red (band 3, 0.64 μm), green (band 2, 0.51 μm) and blue (band
1, 0.47 μm). Finally, the daily liquid water path (LWP) observations from the MODIS
Aqua Collection 6 Level-3 production are used to evaluate the model performance on
simulating cloud water.
**2.3 Model evaluation**
Comparisons between the simulated and observed meteorological parameters from
26 October to 2 December 2018 in Nanjing are shown in Fig. 2. The model can
reproduce the temporal variation of observed meteorological variables, such as T2, RH,
WS10, and WD10, with the relatively high correlations of 0.89, 0.68, 0.47 and 0.55,
and small root-mean-square errors (RMSEs) of 1.7 °C, 9.7 %, 0.6 m s$^{-1}$ and 61.7$^{o}$,
respectively. The simulated T2, RH, and WS10 are slightly lower than observations,
with the mean biases of -0.4 °C, -1.4 %, and -0.1 m s$^{-1}$, respectively (Table 2). There
was almost no precipitation during this period. Similarly, the simulated precipitation is
also quite limited except for the date on 2 December. Overall, the simulated
meteorological fields are reasonable in Nanjing.
**3   Results and discussions**
**3.1 Chemical simulations**
From 26 November to 2 December 2018, Nanjing and its surrounding cities



suffered from a severe haze-fog event for seven days (fog areas are shown in Fig. S1).
The average $PM_{2.5}$ concentrations and RH in Nanjing exceeded 115 μg m$^{-3}$ and 85%,
respectively, and the visibility is less than 50 meters in some areas.

The hourly and diurnal variations of simulated and observed $SO_2$, $NO_2$, $NH_3$,

$HNO_3$, and HONO as well as SNA and $PM_{2.5}$ concentrations are shown in Fig. 3 and 4.
The magnitudes and temporal variations of air pollutants from the simulations and
observations are generally consistent. However, the model overestimates $SO_2$ by 114 %
and underestimates sulfate by over 80 %, and thus underestimates the sulfur oxidation
ratio (SOR) by 81 %. A low oxidation rate of $SO_2$ to sulfate in the model has been found
in previous studies (Gao et al., 2018). Possible explanations are probably associated
with unclear or imperfect chemical mechanisms of sulfate formation in the models.
(Moch et al., 2018; Sha et al., 2019; Shao et al., 2019). Additionally, it is noted that the
observed sulfate concentration has an obvious diurnal cycle with the peak occurring at
noon, corresponding to the timing of fog dissipation. Sulfate mass concentration can
remain at a relatively high level in fog water during the night and early morning due to
the contribution from aqueous chemistry, inducing a significant increase of sulfate
when fog droplets evaporate at noon (Xue et al., 2016). However, the simulated sulfate
shows a flatter diurnal cycle, with a much smaller concentration enhancement rate (0.45
μg m$^{-3}$ hr$^{-1}$) from early morning to noon compared to the observations (2.3 μg m$^{-3}$ hr$^{-1}$),
suggesting that model possibly underestimates the formation of sulfate via aqueous-
phase chemistry in clouds.

Globally, the aqueous sulfate formation is mainly from the oxidation of S(IV) by

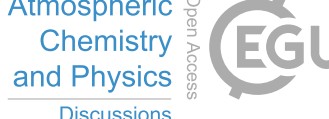

$H_2O_2$ and $O_3$, and almost 50% from the oxidation by $H_2O_2$. Previous studies indicated
that the heavy pollution in China is usually associated with a weak photochemical
activity, while the formation of atmospheric oxidant species (e.g. OH, $H_2O_2$, and $O_3$) is
driven by photolysis, which could suppress the formation of sulfate via the oxidation
of S(IV) by $H_2O_2$ and $O_3$ during the haze-fog events (Xue et al., 2016; Li et al., 2017a;
Wang et al., 2020; Liu et al., 2020b). Therefore, the aqueous-phase oxidations of S(IV)
by $NO_2$ and $O_2$ (TMI-catalyzed) could play an important role in sulfate formation. It is
noted that the observed HONO concentrations rise remarkably at noon, which is quite
consistent with the diurnal cycle of sulfate (Fig. 4(b, g)), while most of the HONO is
produced via $SO_2$ oxidation by $NO_2$ in aqueous phase according to previous studies
(Liu et al., 2019). It is therefore suggested that the aqueous-phase oxidations of S(IV)
by $NO_2$ is possibly the main pathway of sulfate formation during this haze-fog event.
However, the simulated HONO is almost an order of magnitude lower than the
observations and has no obvious diurnal variations as shown in the observations.
Although the diurnal pattern of $NO_2$ is consistent in the model and observations,
and the averaged NMB is only 12 %, the nitrate concentrations are 184 % higher in the
model than in the observations, especially at night, suggesting that the model
overestimates the nitrate nocturnal formation pathway, that is, the $N_2O_5$ heterogeneous
hydrolysis uptake on the surfaces of deliquescence aerosols (Lowe et al., 2015; Brown
et al., 2016; Chang et al., 2016). The relatively high $N_2O_5$ uptake coefficient ($\gamma_{N2O5}$) and
missing of heterogeneous production of nitryl chloride ($ClNO_2$) from the $N_2O_5$ uptake
on chloride aerosols in the model, lead to the overestimation of the simulated nitrate



mass concentration (Sarwar et al., 2012, 2014; McDuffie et al., 2018). Besides,
overestimations of the $HNO_3$ and nitrate ($TNO_3 = HNO_3 + NO_3^-$) concentrations in the
model are also caused by the insufficient removal of $TNO_3$. Therefore, too much
$TNO_3$ may consume a large amount of $NH_3$ to a certain extent, further inhibit the
sulfate formation.
The molar concentrations of total ammonium ($TNH_4 = NH_3 + NH_4^+$) are
generally consistent in the simulations and observations, i.e. 2.1 mol m$^{-3}$ in the
simulation and 2.5 mol m$^{-3}$ in the observation, but the simulated $NH_3$ is 91 % lower and
ammonium is 57 % higher than the observations (Fig. 3(e, f)). This is partly due to the
overestimation of $TNO_3$ in the model (Wang et al., 2013). On the other hand, aerosol
acidity is a key factor driving the semi-volatile partitioning of aerosol species, and
lower aerosol pH is conducive to the existence of ammonium in the particle phase. As
shown in Fig. S2, the model underestimates aerosol pH by 0.8, which leads to the
discrepancies of $TNH_4$ gas-particle partitioning.
The simulated $PM_{2.5}$ concentrations are significantly higher than the observations
(twice during daytime and three times during night). As CBMZ-MOSAIC only predicts
primary organic species but does not consider the formation of secondary organic
aerosol, the organic mass concentration must assumedly be underestimated in the model.
Therefore, the overestimation of $PM_{2.5}$ is mainly due to the overestimation of SNA,
namely nitrate and ammonium. Additionally, the overestimation of primary inorganic
aerosols mass concentrations in the model can also lead to a positive bias of $PM_{2.5}$.
**3.2 Cloud water**





Based on the above analysis, we speculated that underestimation of sulfate in the
model is due to the insufficient in-cloud aqueous-phase formation and/or missing
mechanisms in the model. The cloud water is the most uncertain factor to modulate in-
cloud aqueous-phase chemistry (Ervens et al., 2015; Xie et al., 2018). Therefore, it is
necessary to evaluate the simulated cloud water in the model.
Figure 5 shows the spatial distribution of simulated fog from 26 November to 2
December over YRD. The fog area was identified once LWP is above a threshold of 2
g m$^{-2}$ (Jia et al., 2019). The model can generally reproduce the distribution
characteristics of the fog area observed at 08:00 every day during this period, except
for the date on 27 November (the observed fog areas are shown in Fig. S1).
The LWC at the lowest level of the model has an important impact on the SNA
formation at surface. LWC was not observed simultaneously during this period, so
visibility (VIS) is usually used to assess the simulated LWC as it is a function of LWC
and cloud droplet number ($N_c$) (Eq. (1); Gultepe et al., 2006).
$VIS[m] = 1002/(LWC[g\,cm^{-3}] \times N_c[cm^{-3}]^{0.6473})$           (1)
Figure 6 compares the spatial distribution of VIS from simulations and observations
(threshold of VIS < 1000 m). The simulated VIS has similar spatial pattern and
magnitude with the observed VIS. However, the model tends to overestimate VIS,
especially on 27 November, likely because the LWC is underestimated. The
underestimation of LWC during this period may be related to the bulk microphysical
scheme used in the model (Khain et al., 2009; Jia et al., 2019).
To quantitatively evaluate the modeled cloud water, we compared the simulated



LWP with the MODIS daily observation (Fig. 7). The model can reproduce the spatial
distribution of observed LWP but somewhat underestimates LWP in some areas, e.g.
Jiangsu Province. Comparisons of the cumulative probability distribution of the
simulated and observed LWP are shown in Fig. 8. The probability distribution of the
simulated LWP is mainly concentrated in the lower LWP, e.g. the probability of the
simulated LWP less than 20 g m$^{-2}$ is ∼ 80 %, while the observed one is only 30 % (Table
2). The modeled probabilities are 49 % lower than the observed ones for larger LWP (>
20 g m$^{-2}$). The results are consistent with previous studies (Mueller et al., 2006; Kay et
al., 2012; Wang et al., 2013; Sha et al., 2019).
As stated above, the model underestimates the sulfate mass concentration and
cloud water simultaneously during the haze-fog event. The underestimation of cloud
water possibly leads to the insufficient contribution of in-cloud aqueous-phase
chemistry to sulfate formation, which could explain the underestimation of sulfate
during the haze episode, but has been overlooked by most previous studies. Therefore,
the next section uses the observed LWP from MODIS to constrain the simulations and
explore the impact of cloud water on SNA simulation.
**3.3 Sensitivity experiments**
**3.3.1 Constrain of cloud water in the model**
The logarithmic function is used to fit the cumulative probability distributions
(CPD) for both the observed and simulated LWP (Fig. 8) values. The corresponding
equations of the fitting are:
$F_o = -6.4 + 16.5\ln(x_o + 1.0)$                           $(0 \leq x_o \leq 500 \, \text{g m}^{-2})$ (2)



$F_{\mathrm{m}} = 59.1 + 6.7\ln(x_{\mathrm{m}} + 5.8)$                    $(0 \leq x_{\mathrm{m}} \leq 500\ \mathrm{g\ m^{-2}})$ (3)
Where subscripts o and m represent the observation and model, while $F$ and $x$ represent
CPD and LWP. To update the modeled LWP with satellite observations, we use the
histogram matching method (Richard, 2013), so that the CPD function of the simulated
LWP after constraining is the same as the observations, i.e., $F_{\mathrm{m}}^{\mathrm{c}} = F_{\mathrm{o}}$. Consequently,
the equation for transforming the modeled LWP is:
$x_{\mathrm{m}}^{\mathrm{c}} = 53.0 \times (x_{\mathrm{m}} + 5.8)^{0.4} - 1$                    (4)
Where the subscript c presents the correction with MODIS observations.

We apply the Eq. (4) to modify the cloud water in the aqueous chemistry module

only while cloud water amount in other modules (i.e. microphysics, cumulus
parameterization, wet scavenging, and radiative transfer modules) remain unchanged
to ensure that other physical and chemical processes are self-consistent between the
control and sensitivity model simulations. This sensitivity experiment is called Sen_c.
Consequently, the changes don't affect the cloud properties used in the radiative transfer
calculations. As such, gas phase production rates are intact. However, cloud-induced
changes in aqueous phase production do alter the mixing ratios of $SO_2$ and other
oxidants (e.g., OH and $H_2O_2$), which could in turn impact the rate of gas phase oxidation.
In addition, the changes in cloud water can affect the production rates of sulfate by
changing the hydrogen ions concentrations ($[H^+]$). The pH of cloud water is considered
as one of the important parameters affecting the aqueous-phase reaction rates. As shown
in Fig. S3, constraining the simulated cloud water alone results in a decrease of cloud
water pH (2.4) during this period. To eliminate the influence of changes in cloud water



pH (from MODIS-based change of cloud water) on the sulfate production, we also
increase the cloud water pH by 2 in another sensitivity experiment (Sen_c_pH) to make
cloud water pH as close as possible to the control simulation. The experiment
descriptions are shown in Table 3.
**3.3.2 Impact of cloud constraint on SNA**
Figure 9 shows the spatial distribution of the simulated SNA in the control and
sensitivity simulations, as well as the difference between the two simulations. The
simulated sulfate concentration in Sen_c_pH is 6 $\mu g\ m^{-3}$ larger than the Control over
the entire YRD, with the biggest difference in the south of Jiangsu and the east of Anhui
province, corresponding to the area mostly affected by this haze-fog event (Fig. S1). It
is indicated that corrected cloud water increases the contribution of the aqueous-phase
chemistry to sulfate formation, thereby reducing the negative bias of simulated sulfate.
The formation of sulfate greatly limits the nitrate production, so the simulated nitrate
in Sen_c_pH is decreased by 35 $\mu g\ m^{-3}$ compared to the Control over the entire YRD.
However, the ammonium simulated by Sen_c_pH is larger than the results of Control
run in most areas of YRD, with the average difference of 9 $\mu g\ m^{-3}$. As the inorganic
aerosol system is essentially an acid-base titration, an increase in S(VI) concentration
can neutralize more $NH_3$ to form ammonium sulfate (($NH_4$)$_2$$SO_4$) or ammonium
bisulfate ($NH_4HSO_4$), leading to an increase of simulated ammonium concentrations.
As shown in Fig. 10 and Fig. 11, Sen_c_pH significantly improves the simulation
of sulfate, i.e. increases sulfate by 11.8 $\mu g\ m^{-3}$ (295 %), and decreases NMB by 53.5 %.
Also, the simulation using corrected cloud water can reproduce the diurnal cycle and





capture the peak concentration of sulfate at noon, with the concentration increased rate
of 1.8 $\mu$g m$^{-3}$ hr$^{-1}$ from early morning to noon, which is not seen in the Control run.
Meanwhile, Sen_c_pH decreases the absolute bias of the simulated nitrate from 184.0%
(Control) to 50.1 %, and greatly reduces the nitrate concentration at night, and thus
predicts a better diurnal cycle. However, the simulation with corrected cloud water
leads to a minor increase of ammonium.

Overall, the simulation with MODIS-corrected cloud water can obviously

decrease the model bias of SNA to 14.1 % from 77.2 % in Control run (Fig. 11(a)). The
proportion of sulfate in SNA also significantly increases from 2.5 % (Control) to 20.2 %
(Sen_c_pH), which is much close to the observation (23.9 %), but still 6 $\mu$g m$^{-3}$ lower
than the observations. A few possibilities can explain the discrepancies. The model
possibly underestimates the cloud water pH, with the value of 3.3 in Sen_c_pH (Fig.
S4), which is relatively lower than the global typical cloud/fog water pH of 3-6 and the
mean value of 4-6 suggested by Pye et al. (2020). The observed fog water pH in Nanjing
from previous studies (Li et al., 2008; Lu et al., 2010; Qin et al., 2011; Yan et al., 2013;
Hong et al., 2019) are summarized in Table 4, suggesting that the fog water pH in
Nanjing is generally between 4.3 and 6.5. Therefore, the relatively lower fog water pH
simulated by the model could limit the aqueous-phase formation of sulfate to some
extent. Note that the aqueous-phase oxidation of S(IV) by NO$_2$ requires the cloud water
pH of about 6, thus the more acidic cloud water in the model is not conducive to this
reaction. Moreover, the model lacks SO$_2$ heterogeneous reactions on aerosol water (Li
et al., 2017a; Shao et al., 2019) and other aqueous-phase reactions in clouds, such as



the aqueous oxidation of S(IV) by HCHO and hydroxyl hydroperoxide (ISOPOOH) to
form hydroxy-methane sulfonate (HMS) and sulfate (Moch et al., 2018; Dovrou et al.,
2019), can also explain the sulfate underestimation even though the cloud water has
already been corrected. In addition, cloud constraints are based on the MODIS LWP,
which has been reported with an uncertainty range of ±30 % (Dong et al., 2008; Min et
al., 2012; Khanal et al., 2018).

It should also be noted that compared to the observations, Sen_c_pH

underestimates nitrate and overestimates ammonium in SNA (Fig. 11(b)), which can be
ascribed to the underestimation of atmospheric acidity in the model, including the pH
of aerosol and cloud/fog water. The hydrogen ion activity in aqueous aerosols can affect
the partitioning of $TNO_3$ and $TNH_4$ between the gas and aerosol phases. Lower
aerosol pH favors partitioning of $TNO_3$ toward gaseous $HNO_3$ rather than aerosol
nitrate. In contrast, $TNH_4$ partitions toward gaseous $NH_3$ at higher aerosol pH (Weber
et al., 2016). The simulated aerosol pH in Sen_c_pH is lower than the observations (Fig.
S2), which is not conducive to the existence of aerosol nitrate. Additionally, because
the scavenging efficiency of $TNO_3$ and $TNH_4$ is dependent upon cloud water pH, the
acidic cloud water in the model can also cause these discrepancies.

**4    Conclusions**

Accurately predicting the concentrations and chemical components of particulate

matter are still very challenging for climate and air quality models. In this study, we
evaluated the WRF-Chem performance on simulating inorganic aerosol components of
$PM_{2.5}$ during a haze-fog event in Nanjing, and investigate the possible reasons





contributing to model bias in simulating SNA compared with the observations.

Our results presented that WRF-Chem overestimates $SO_2$ by 114 %,

underestimates sulfate by 81%, and fails to reproduce the diurnal cycle of sulfate, i.e.
the peak concentration at noon, which corresponds to the timing of fog dissipation. In
contrast, the model bias of $NO_2$ is much smaller (NMB = 12 %), but the nitrate is
overestimated by 184 %, especially its nocturnal concentration. Although the molar
concentrations of total ammonium are generally consistent in the simulations and
observations, the model underestimates $NH_3$ by 91 % and overestimates ammonium by

57%.

The underestimation of sulfate concentration is consistent with previous findings.

However, our work stands in contrast to previous studies that adding $SO_2$ heterogeneous
mechanism in the model to improve the simulation of sulfate. Cloud/fog droplets are
the important reactors in which dissolved $SO_2$ reacts with oxidations to form sulfate,
but the model underestimates cloud water (both surface LWC and LWP) simultaneously.
Therefore, the cloud water in the model was constrained based on the MODIS LWP
observations, and sensitivity experiments were conducted to explore the impact of
corrected cloud water on SNA simulation. Compare with control run, the simulation
with MODIS-corrected cloud water significantly improves the simulation of sulfate, i.e.
increases the concentration by nearly 3 times and decreases NMB by 53.5 %, as well
as reproduces the diurnal cycles. Additionally, corrected cloud water decreases the bias
of simulated nitrate by 134 %, especially the nocturnal concentrations, thus predicting
a better diurnal cycle. Although the simulated ammonium is higher than the control



simulation and observation, corrected cloud water decreases the model bias of SNA to
14.1 % from 77.2 % (Control).
However, even after the MODIS-based adjustment of cloud water, the simulated
sulfate is still 6 μg m$^{-3}$ (27.5%) lower than the observations, suggesting that the model
possibly underestimates the cloud water pH (the value of 3.3), which is not conducive
to the in-cloud aqueous-phase oxidation of S(IV) by NO$_2$. Missing of SO$_2$
heterogeneous reactions on aerosol water (e.g., TMI-catalyzed oxidation) and other in-
cloud aqueous-phase reactions (e.g., S(IV) oxidation by HCHO and ISOPOOH) in the
model can also lead to underestimating the sulfate concentrations. In addition, the
constraints of cloud water are based on the MODIS observations, which are themselves
subject to retrieval uncertainties.
The above results emphasize the critical role of cloud water in simulating SNA,
and provide a new perspective on the causes of sulfate underestimation discussed by
the previous studies. More studies are still needed to comprehensively evaluate the
modeled cloud fields to improve the haze prediction in the future.

**Code and data availability:** Some of the data repositories have been listed in Sect. 2.
The other data, model outputs and codes can be accessed by contacting Tong Sha via
shat@nuist.edu.cn.

**Author contributions:** TS performed the model simulation, data analysis and paper
writing. XM proposed the idea, supervised this work and revised the paper. JW gave



scientific suggestions and also contributed to the paper revision. RT processed the
observation data. JZ offered help with the model simulation. FC and YZ provided the
observation data at the NUIST site.

**Competing interests:** The authors declare that they have no conflict of interest.

**Acknowledgment:** This study is supported by the National Natural Science Foundation
of China grants (41675004 & 41975002), the National Key R&D Program of China
grants (2019YFA0606802 & 2016YFA0600404), and the Postgraduate Research &
Practice Innovation Program of Jiangsu Province (grant no. KYCX20_0919). Jun
Wang's participation of this project is made possible through in-kind support from the
University of Iowa.



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





Table 1. Comparison of the simulated and observed meteorological parameters. (T2: 2
meters temperature (°C), RH2: 2 meters relative humidity (%), WS10: 10 meters wind
speed (m·s$^{-1}$), WS10: 10 meters wind speed (m·s$^{-1}$)).

| Variables | Obs | Mod | R | MB | RMSE |
|-----------|------|-------|------|------|------|
| T2 | 11.5 | 11.1 | 0.89 | -0.4 | 1.7 |
| RH2 | 89.9 | 88.5 | 0.68 | -1.4 | 9.7 |
| WS10 | 1.6 | 1.5 | 0.47 | -0.1 | 0.6 |
| WD10 | 134.2 | 138.4 | 0.55 | 4.2 | 61.7 |







Table 2 Statistics of the cumulative probability distribution of observed and simulated
LWP.

| Probability (%) | Observation | Simulation |
|---|---|---|
| 0-20 g m$^{-2}$ | 30 | 79 |
| 20-40 g m$^{-2}$ | 19 | 4 |
| 40-60 g m$^{-2}$ | 15 | 3 |
| 60-80 g m$^{-2}$ | 4 | 2 |
| 80-100 g m$^{-2}$ | 4 | 2 |
| > 100 g m$^{-2}$ | 28 | 10 |







Table 3. Descriptions of the model simulations.

| Experiment name | Description |
| --- | --- |
| Control | Control simulation. |
| Sen_c | Only constrain the simulated LWP according to Eq. (4). |
| Sen_c_pH | Constrain the simulated LWP according to Eq. (4) and increase the cloud water pH by 2. |





Table 4 Summaries of the observed fog water pH during the fog events in Nanjing,
China.

| Study time | pH in fog | Reference |
|---|---|---|
| December 2006 | 5.6 | Li et al., 2008 |
| December 2006 and December 2007 | 5.9 | Lu et al., 2010 |
| December 2007 | 5.5 | Qin et al., 2011 |
| December 2009 | 6.0 (radiation fog) 5.6 (advection radiation fog) 4.3 (advection fog) | Yan et al., 2013 |
| November 2016 to January 2017 | avg: 5.7, min: 5.0, max: 6.5 | Hong et al., 2019 |



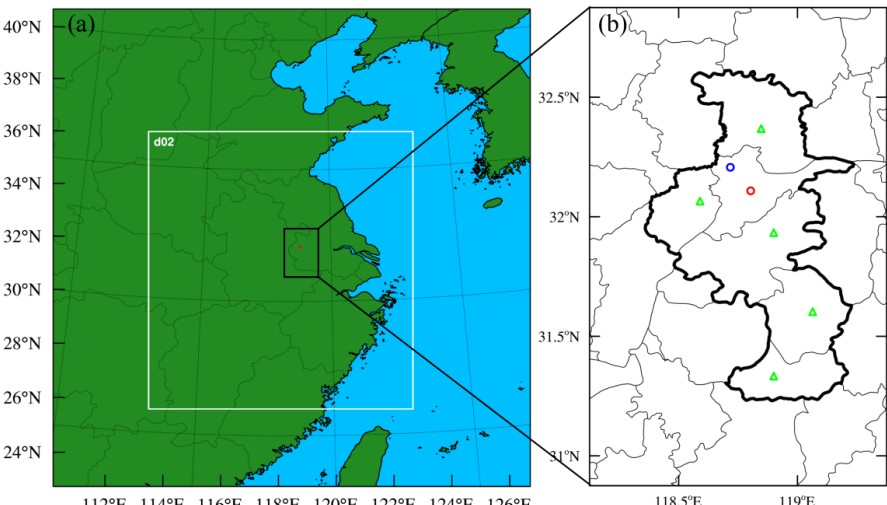


Figure 1. (a) The model domain (Solid red dot is Nanjing). (b) The location of sites with
in-situ measurements on meteorological variables and air pollutants (Green triangles,
red and blue circle denote the routine meteorological stations, Maigaoqiao air quality
monitoring site, and Nanjing University of Information Science & Technology
(NUIST), respectively).

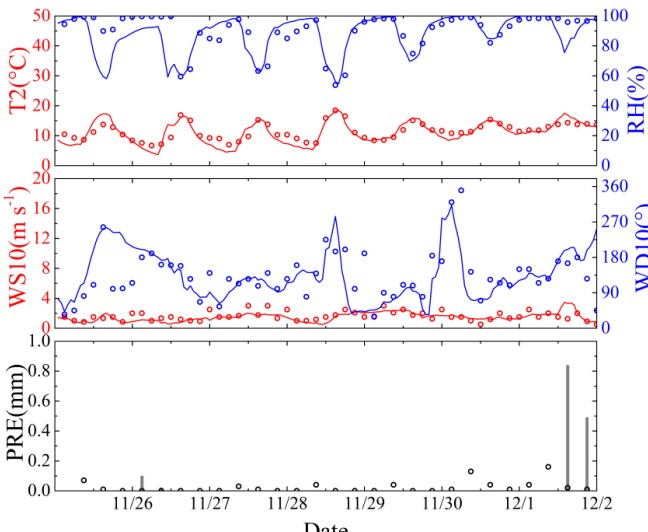

Figure 2. The performance of the simulated hourly meteorological parameters (2m

temperature (T2), 2m relative humidity (RH), 10m wind speed (WS10), 10m wind

direction (WD10), and 6 h accumulation precipitation (PRE)) during the haze-fog event

in Nanjing. Scatters and solid lines (or columns) represent observations and simulations,

respectively.

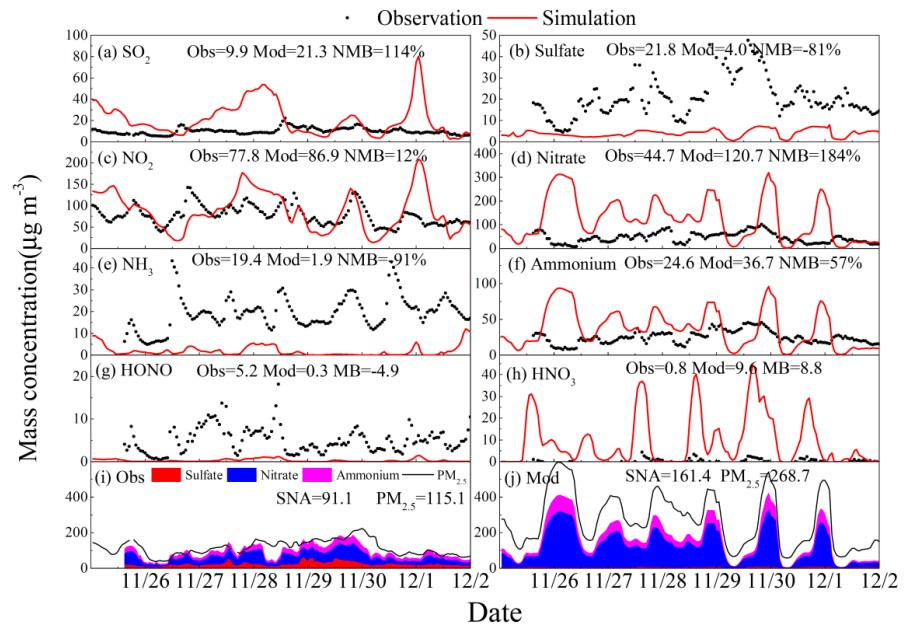

Figure 3. Time series of the simulated and observed hourly gas precursors ((a) $SO_2$, (c)

$NO_2$, (e) $NH_3$, (g) HONO, (h) $HNO_3$), as well as (b) sulfate, (d) nitrate and (f)

ammonium concentrations. The stacked diagram of hourly SNA and $PM_{2.5}$

concentrations from (i) observations and (j) simulations during the haze-fog event in

Nanjing.



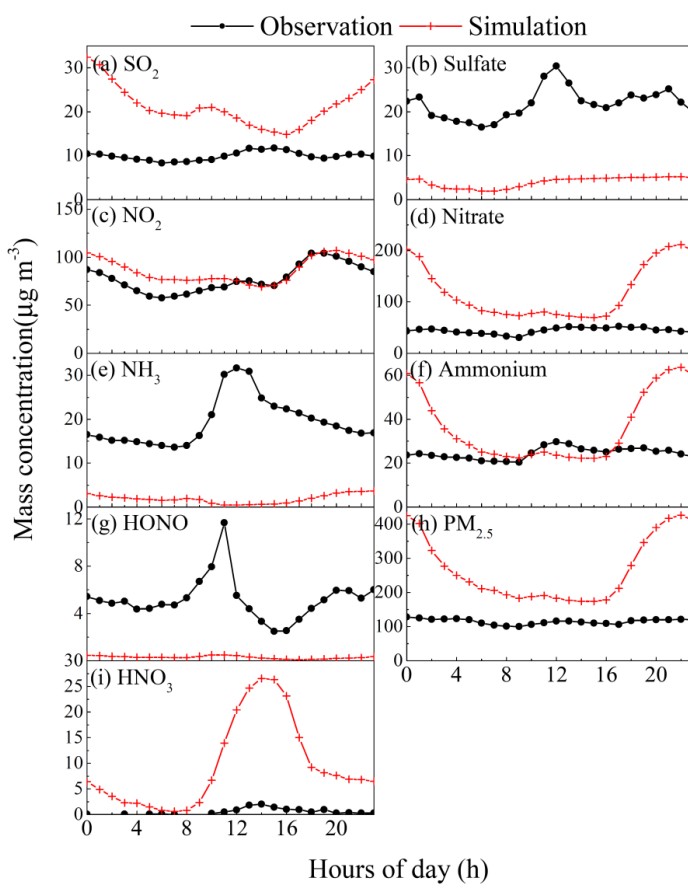

850

Figure 4. Diurnal cycles of the simulated and observed mass concentrations of gas

precursors ((a) SO$_2$, (c) NO$_2$, (e) NH$_3$, (g) HONO, (i) HNO$_3$), as well as (b) sulfate, (d)

nitrate, (f) ammonium and (h) PM$_{2.5}$ averaged during the haze-fog event in Nanjing.

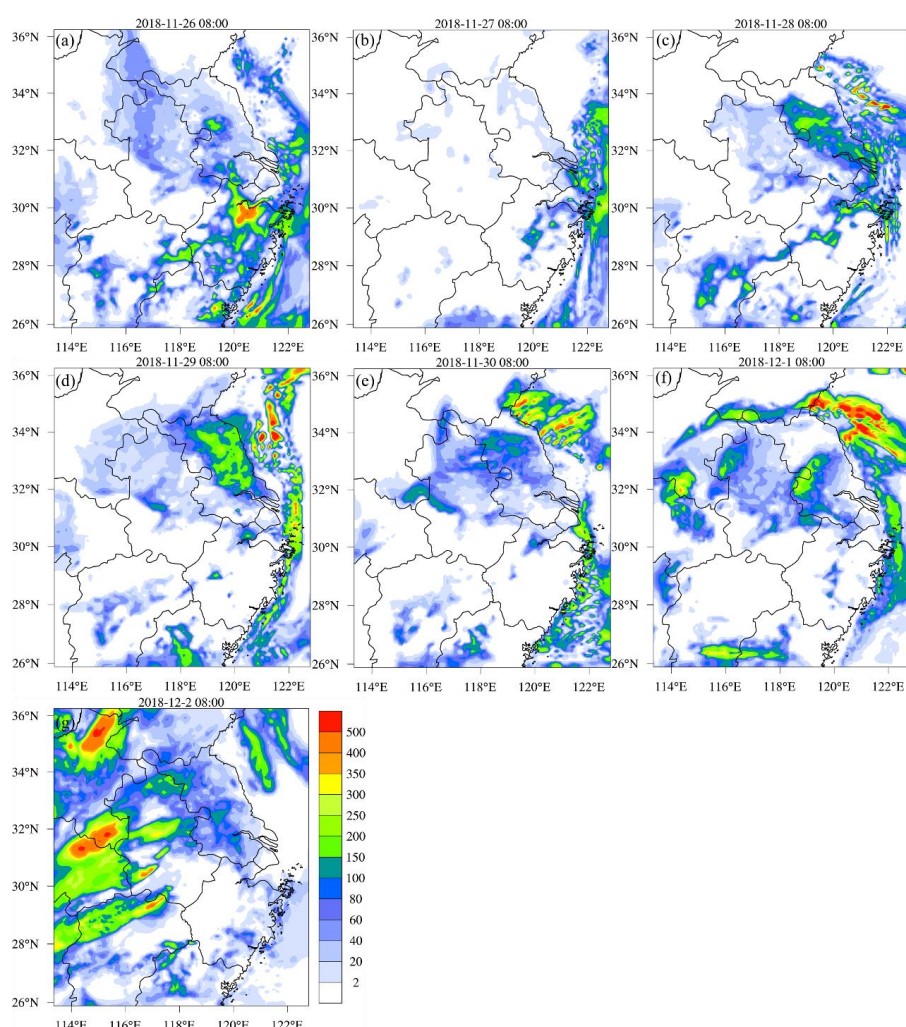

854

Figure 5. Distribution of the simulated liquid water path (LWP, unit: g m$^{-2}$) at 08:00

from 26 November to 2 December over YRD.

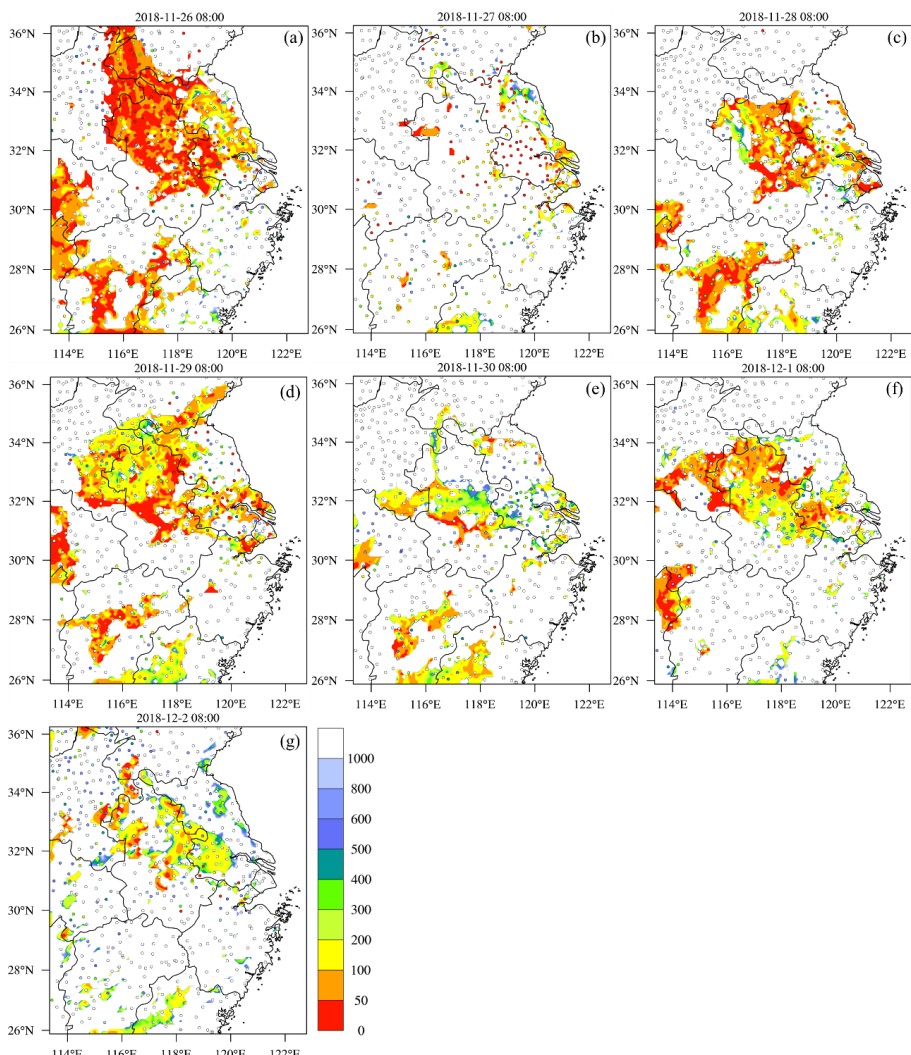

Figure 6. Distribution of the simulated and observed visibility (unit: m) at 08:00 from

26 November to 2 December over YRD. The circles represent the MICAPS

observations.

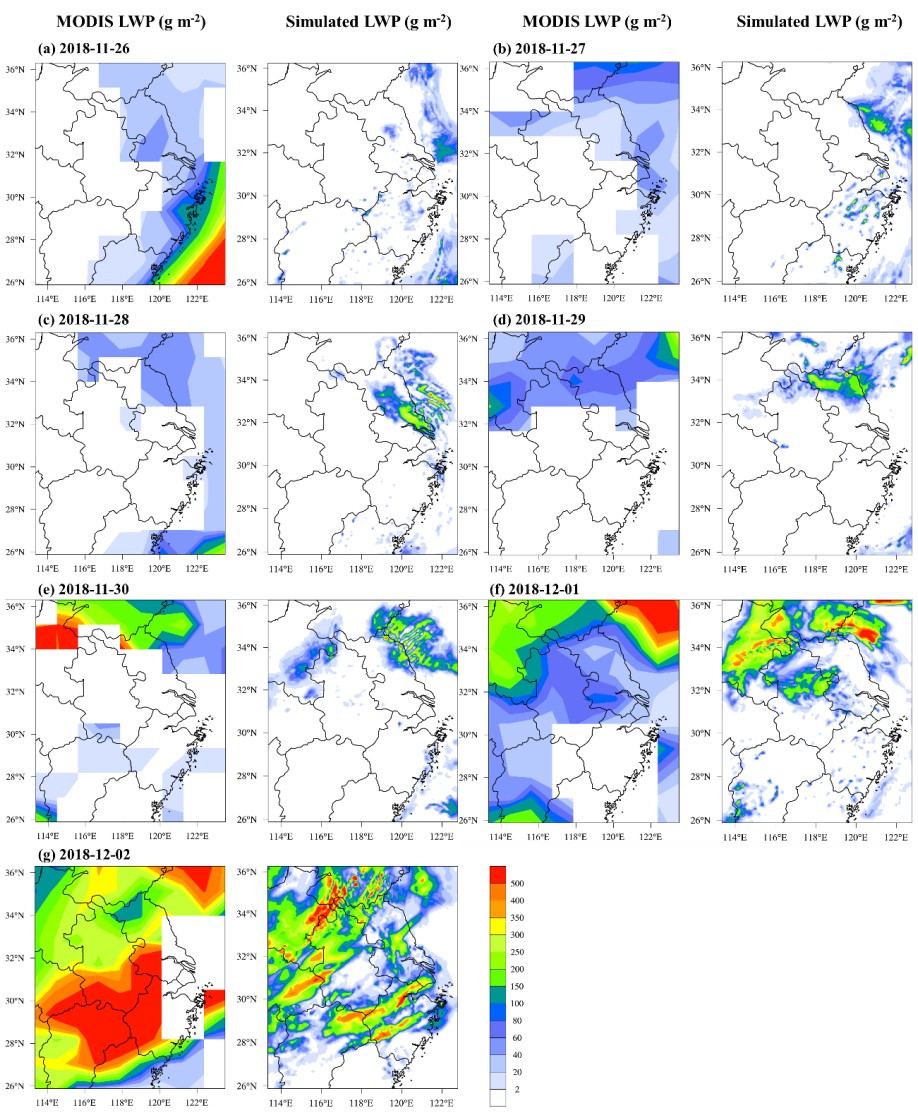

862

Figure 7. Distribution of LWP (unit: g m⁻²) from the MODIS observations (columns 1

and 3) and simulations (columns 2 and 4) at 13:30 from 26 November to 2 December

over YRD.

866





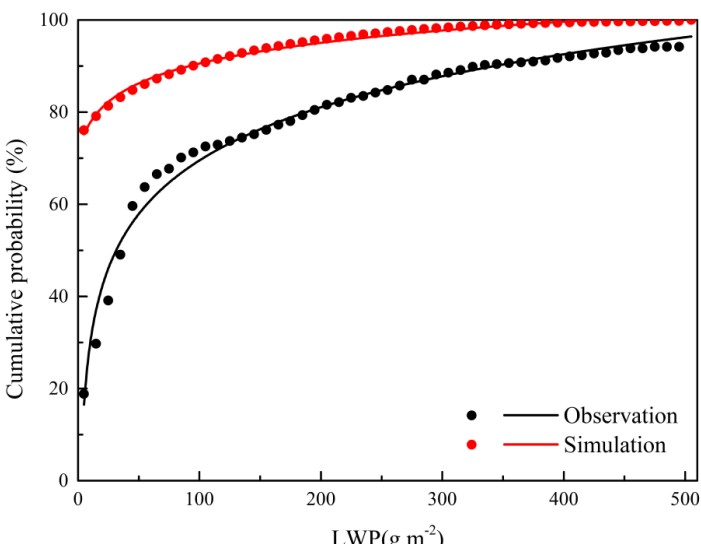

867

Figure 8. The cumulative probability distribution of LWP between the MODIS

observations and simulations. Results are based on statistics of the observed and

simulated daily LWP during the haze-fog event over YRD. The lines are the fitting

functions.

872



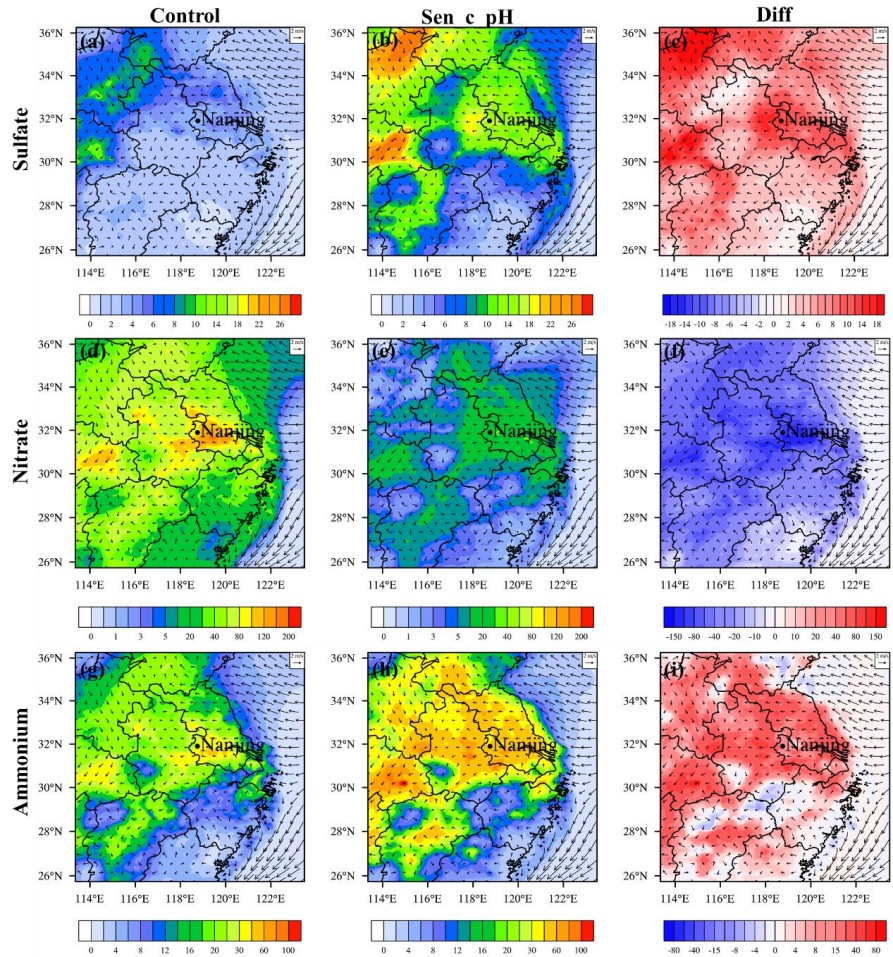

873

Figure 9. Distribution of the simulated sulfate, nitrate, and ammonium (SNA) in the

Control run (a, d, g) and Sen_c_pH (b, e, h) simulation, and the differences of simulated

SNA between the two simulations (c, f, i) during the haze-fog event over YRD. The

black arrows indicate the simulated surface wind fields.

878



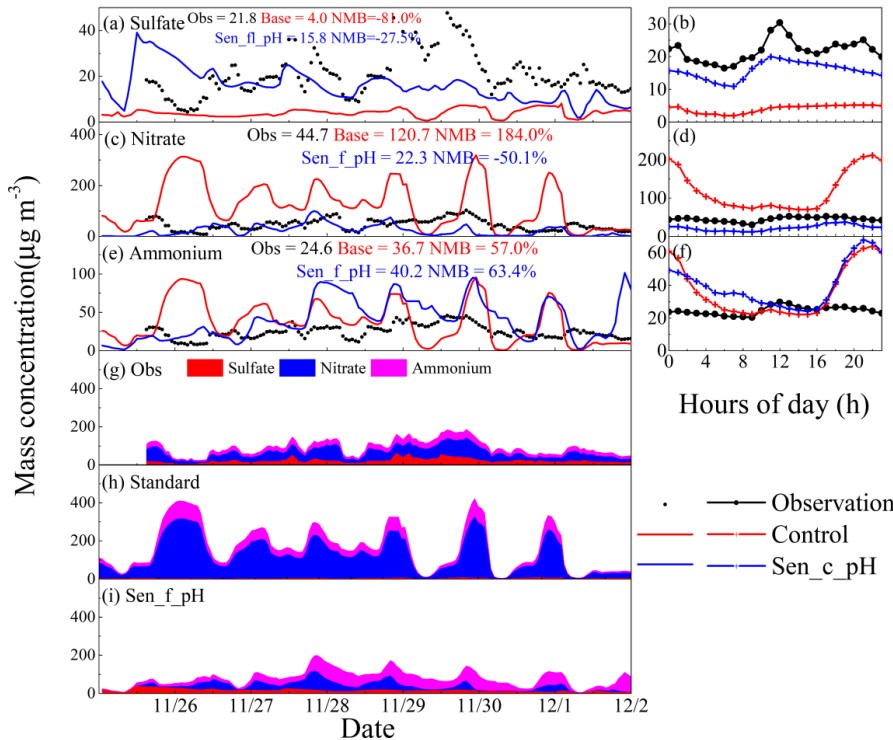

Figure 10. The hourly and diurnal variations of simulated (Control and Sen_c_pH) and

observed (a, b) sulfate, (c, d) nitrate, and (e, f) ammonium concentrations. The stacked

diagram of hourly SNA concentrations from (g) observations, (h) Control run, and (i)

Sen_c_pH simulations during the haze-fog event in Nanjing.





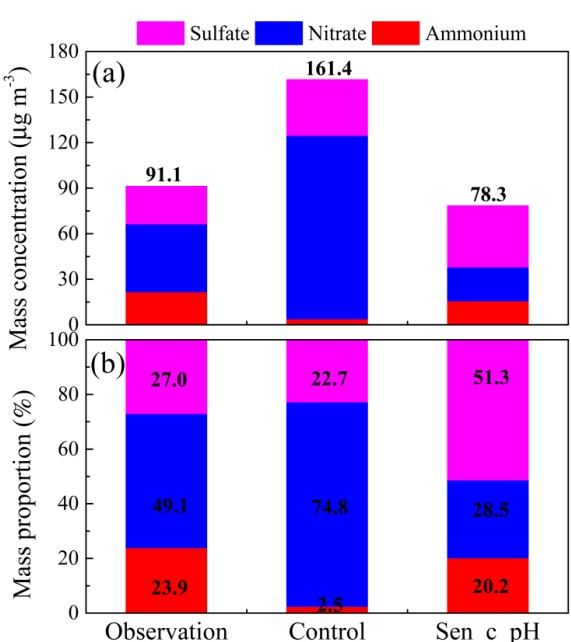


Figure 11. (a) The average mass concentrations and (b) proportion of the observed and
simulated (Control and Sen_c_pH) SNA during the haze-fog event in Nanjing.