# Peer review of "Improvement of inorganic aerosol component in PM₂.₅ by constraining aqueous-phase formation of sulfate in cloud with satellite retrievals: WRF-Chem simulations"

_Atmospheric Chemistry and Physics, 2020_

## Referee Comment (RC1) · Anonymous Referee #1 · 24 Sep 2020

This paper presents aerosol predictions at Nanjing, China during a haze fog event in which high concentrations of sulfate were observed. The default version of the model underestimated sulfate concentrations and the authors hypothesize that underpredictions in cloud water was a likely explanation. If simulated cloud water is too low it is reasonable to assume the amount of sulfate produced by aqueous chemistry pathways would be reduced. The authors crudely adjust the simulated cloud water to show that sulfate predictions increase when cloud water is increased. This is an obvious result; however, they are correct to note that most air quality studies do not investigate this

pathway. Nevertheless, there are flaws in their methodology and analysis that need to be corrected before the paper can be suitable for publication. In addition, there are numerous grammatical errors that make the paper difficult to read. I tried to present suggestions in the abstract, but did not provide corrections for the entire manuscript. The authors should find some assistance to improve the readability of the manuscript.

Specific Comments:

Line 21: Change "it is still a big challenge" to "it is still challenging" or "it is still very challenging".

Line 22: Change "in the numerical model" to "in numerical models"

Line 24: Change "in WRF-Chem" to "in the WRF-Chem model". Change "(November 2018)" to "during November 2018".

Line 25: Change "the sulfate" to "sulfate"

Line 28: Change "dissipation, suggesting that the model" to "dissipation that suggests the model"

Line 30: Change "184 % and 57 %" to "184% and 57%" and similarly elsewhere in the paper.

Lines 30-31: Change "These ultimately result in the simulated SNA 77.2 % higher than the observations." to "These overestimates contribute to the simulated SNA being 77.2% higher than observed."

Line 31-32: Change "However, as the important aqueous-phase reactors, cloud water are simultaneously underestimated by the model." to "However, cloud water is also underestimated by the model which is a pathway for important aqueous-phase reactions."

Line 33: Change "Therefore, the modeled cloud water was constrained" to "Therefore, we constrained the simulated cloud water in a sensitivity simulation"

Line 36: Change "and can reproduce its diurnal cycles, i.e. the peak concentration at noon" to "and reproduces its diurnal cycle with the peak concentration occurring at noon"

Lines 36-39: This sentence quotes a number which presumably is an average, but then notes differences during nighttime which is awkward. The sentence needs to be reworded.

Line 39: Change "the observation to "the observations"

Line 40: Change "lead" to "leads". Change "of SNA" to "in SNA"

Line 41: Change "the cloud water" to "cloud water content"

Line 42: Change "bias of SNA simulation" to "simulated SNA bias".

Line 43: Change "of cloud water can lead to model bias" to "in cloud water content can contribute to model biases"

Lines 43-44: This result is not very surprising but a useful exercise since most air quality simulations tend to focus on uncertainties in emissions, trace gas and aerosol interstitial chemical mechanisms, and boundary layer depth. The effects of clouds are examined less frequently From a process-level perspective, it would be more pleasing to simulate the amount and spatial distribution of cloud water more accurately. That could be achieved in part by constraining the ambient meteorology using data assimilation rather than constraining cloud water content alone.

Lines 113-114: Change "the size distribution" to "droplet size distribution"

Line 133: Change "Model" to "model".

Line 141: Change "with the horizontal resolution" to "with a grid spacing". Grid spacing and resolution are different things. A model can resolve phenomena that are bigger than ∼5 to 6 times the grid spacing.

Line 142: Change "with 9 km" to "with a grid spacing of 9 km"

Line 144: Change "parameterization schemes" to "parameterizations". The two words are redundant. The rest of the sentence switches between "parameterization" or "scheme" for unknown reasons. Pick one and be consistent.

Line 151: Change "and Model" to "and the Model"

Line 159: Change "condition" to "conditions".

Line 162: change "as the" to "used as a"

Lines 171-177: Detail chemistry measurements are only available at two sites, so issues of spatial representativeness need to be address when evaluating the performance of the model. I do not doubt that cloud water is an important factor, but uncertainties in ambient conditions could displace plumes and contribute to model error. The models compares well to surface observations, but there is not assessment of how the model performs aloft which is important as well.

Lines 183-185: State the spatial grid spacing of the observations.

Lines 186-196: While it is encouraging the simulated meteorology is reasonable for one surface site, the authors do not provide the context of what this means in terms of the larger-scale meteorology that is contributing to fog and clouds over a much wider area in which aqueous chemistry is occurring.

Lines 256-257: This is the first place the authors discuss simulated pH. It is not clear in text how they arrived at pH since it is not a standard output of WRF-Chem. In Fig. S2 they state they use an offline ISORROPIA model to compute pH. But that is a different thermodynamic module than the one in MOSAIC, so saying these values are the simulated pH is not correct. So all the assumptions in modeled pH they describe in the manuscript are questionable.

Lines 271-275: The authors describe fog in the text, but the figure is showing LWP

which is a vertically integrated quantity. Thus, LWP may not reflect cloud water at the surface. The text implies all the LWP in Figure 5 is fog, but it is possible that parts of those regions are just clouds. If the authors which to describe the areas in Figure 5 as fog, then some additional explanation is needed or different quantities should be plotted.

Lines 276-286: Now the authors focus on surface LWC which is more appropriate in terms of fog. The note that visibility is overestimated by the model, especially on the 27th. But they fail to note that the model seems to underestimate visibility on other days and in other regions, which would argue against their main hypothesis. The amount of sulfate resulting from aqueous chemistry is likely a multiday process so that what is observed at Nanjing is likely an accumulation of sulfate formed over the region that is advected over Nanjing. Another problem with this comparison is that visibility could just be due to high levels of pollution that may be enhanced by aerosol water – and not fog.

Line 298: Differences in the spatial resolution makes it difficult to really assess the potential errors in the model. It would make more sense to average the simulated LWP over the same area as the MODIS grid cells for a more fair comparison. This would affect the fittings performed in the next section.

Line 318: I understand the rationale for modifying the cloud water only in the aqueous chemistry module. But the authors need to put this into context with their findings later in the paper. In reality, changing water content will ultimately impact other aspects of the coupled processes in the model and ultimately change predictions of aerosol species and aqueous chemistry.

Line 397-398: The authors state they investigated the possible reasons contributing to model bias, when in fact they only investigated two: cloud water and specification of pH. The authors need to be more clear and explicit here. The authors could have strengthened their argument by performing other sensitivity simulations that explore

other uncertainties such as emissions and chemical mechanisms and putting those results into the context of the cloud water sensitivity analysis.

Lines 431-434: It would have been useful to provide some more reflection of what could be done to improve the cloud predictions. For example data assimilation is often used to constrain the ambient meteorology which often leads to improved cloud predictions. That is a tool that can be used more routinely for air quality applications and is more pleasing than brute force adjusting cloud water content. Exploring other microphysics representation is another option.

---

## Referee Comment (RC2) · Anonymous Referee #2 · 4 Nov 2020

General comments: East Asian countries and regions are always suffered from serious air pollutions with rapid economic growth in recent decades. And high level emissions of air pollutants in East Asia could further affect regional air qualities, human health, traffic safeties as well as regional or global climate changes. Observations have revealed that severe and persistent haze pollutions occurred frequently in China during recent years. Although the numerical models could capture the loading levels and temporal-spatial variations of the total PM, most of them could not well simulate their chemical components, especially in heavy pollution episodes. Thus, accurately pre-

dicting the concentrations and chemical components of particulate matter are still very challenging for climate and air quality models. In this study, influence of aqueous-phase chemistry on the formation of near surface sulfate as well as the concentrations of total ammonium is carried out to investigate the importance of this process in some polluted episodes, based on observations and numerical evolutions. Therefore, the topic of this study is interesting and novel to some degrees and the paper has a potential for publication in the journal.

Speicfic comments: 1. Both Abstract and Conclusions should be more concise, instead of only repeating the results. 2. Were the aerosol or trace gases from biomass burning taken into account in the simulations? What is the resolution of the emission inventory (MEIC)? Why the emissions in 2016 were used to assess the pollution episode in 2018? 3. What is the resolution of the Himawari-8 and MODIS data? Is the MODIS resolution accurate enough to evaluate the model? 4. It seems that the simulated ammonium (NH4+) has little improvement when simulated the corrected LWC is used. Why? 5. Was the VIS calculated based on the aerosol and trace gases in the model? If so, then the overestimated VIS in the model could not be used to illustrate the reason why simulated LWC is underestimated. 6. Results in this study states that aqueous-phase chemistry plays a very important role in resulting in sever haze pollution. However, there have many polluted episodes in which inorganic aerosols are also growth sharply in the absent of fogs. The authors should make a brief comparison or statement on these two types of pollutions in Results. 7. Fig. 6 is needed to be re-plotted. The circles in the figure could be drawn in larger sizes. 8. English should be corrected throughout the whole manuscript.

---

## Referee Comment (RC3) · Anonymous Referee #3 · 6 Nov 2020

General comments: The chemical transport model is an important tool for the study of air pollution and emission control. The ability of the model to simulate aerosol and its components is an important standard to evaluate the model This manuscript evaluated the WRF-Chem performance on simulating inorganic aerosol components of PM2.5 during a haze-fog event in Nanjing, and investigate the possible reasons of simulating bias compared with the observations. It found that the strong sensitivity of SNA concentration to the cloud water provides an explanation for the bias of SNA simulation. The topic is of interest and the manuscript is generally well written. There are

several issues that need to be addressed before the manuscript can be accepted for publication.

Specific comments: 1. Because LWP is a vertically integrated quantity. Is the large of MODIS MWP possible due to the thickness of the fog is thicker? I think the effect of the vertical profile simulation can be compared. If there is no observation data, vertical sounding and simulation can be compared. 2. What data quality control did authors do to evaluate the model, especially for Himawari 8 and MODIS? 3. line 256: What is pH observation data used in this study? 4. Lines 348-350: What the influence of NH3 and ammonium concentration by changing pH and LWP? Could you provide more detail? 5. Lines 365-371: It seems that cloud water pH is important to the aqueous-phase reactions rates, and the model underestimated the cloud water pH in this study. Why the pH was change from 4.9 to 2.5 by modifying LWP? And the authors need to clarify how to modify the cloud water only in the aqueous chemistry module in this paper. 6. In my opinion, emissions, meteorological, and chemistry mechanisms are also main factors in air quality model. The author should make more discussion to prove the importance of they investigated: cloud water and pH. 7. Line 815: Describe " WS" twice, change "WS" to "WD". 8. Figure 6 should be re-plottedãÃĆ The circles in the figure could be drawn in larger sizes.

---

## Author Comment (AC1) · 3 Jan 2021

Dear Referees,

Thanks for giving us an opportunity to revise our manuscript entitled "Improvement of inorganic aerosol component in PM$_{2.5}$ by constraining aqueous-phase formation of sulfate in cloud with satellite retrievals: WRF-Chem simulations" (ID: acp-2020-760).

We appreciate your positive and constructive comments. We have studied these comments carefully and make revisions on the manuscript. The point-to-point comments and corresponding responses are attached below.

Thanks again. We look forward to hearing from you soon.

With regards,
Yours sincerely,

Tong Sha, Xiaoyan Ma[*], Jun Wang, Rong Tian, Jianqi Zhao, Fang Cao, Yanling Zhang

Thanks to the referee for his/her very thoughtful suggestions. Below we address the reviewers' comments, with the reviewer comments in black, and our response in blue. We have revised the manuscript accordingly, and mentioned the line number of the **tracked revision**.

**Anonymous Referee #1:**

**General comments:**
This paper presents aerosol predictions at Nanjing, China during a haze fog event in which high concentrations of sulfate were observed. The default version of the model underestimated sulfate concentrations and the authors hypothesize that underpredictions in cloud water was a likely explanation. If simulated cloud water is too low it is reasonable to assume the amount of sulfate produced by aqueous chemistry pathways would be reduced. The authors crudely adjust the simulated cloud water to show that sulfate predictions increase when cloud water is increased. This is an obvious result; however, they are correct to note that most air quality studies do not investigate this pathway. Nevertheless, there are flaws in their methodology and analysis that need to be corrected before the paper can be suitable for publication. In addition, there are numerous grammatical errors that make the paper difficult to read. I tried to present suggestions in the abstract, but did not provide corrections for the entire manuscript. The authors should find some assistance to improve the readability of the manuscript.
Thanks to the reviewer for the comments and suggestions.

**Specific Comments:**
Line 21: Change "it is still a big challenge" to "it is still challenging" or "it is still very challenging".
Corrected, see line 21 in the revised manuscript.

Line 22: Change "in the numerical model" to "in numerical models"
Corrected, see line 22 in the revised manuscript.

Line 24: Change "in WRF-Chem" to "in the WRF-Chem model". Change "(November 2018)" to "during November 2018".
Corrected, see line 24 and 25 in the revised manuscript.

Line 25: Change "the sulfate" to "sulfate
Corrected, see line 26 in the revised manuscript.

Line 28: Change "dissipation, suggesting that the model" to "dissipation that suggests the model"
Corrected, see line 29 in the revised manuscript.

Line 30: Change "184 % and 57 %" to "184% and 57%" and similarly elsewhere in the paper.

Thanks for the suggestion, similar errors in the revised manuscript have been corrected.

Lines 30-31: Change "These ultimately result in the simulated SNA 77.2 % higher than the observations." to "These overestimates contribute to the simulated SNA being 77.2% higher than observed."
Corrected, see lines 31-32 in the revised manuscript.

Line 31-32: Change "However, as the important aqueous-phase reactors, cloud water are simultaneously underestimated by the model." to "However, cloud water is also underestimated by the model which is a pathway for important aqueous-phase reactions."
Corrected, see lines 32-35 in the revised manuscript.

Line 33: Change "Therefore, the modeled cloud water was constrained" to "Therefore, we constrained the simulated cloud water in a sensitivity simulation"
Corrected, see lines 35-36 in the revised manuscript.

Line 36: Change "and can reproduce its diurnal cycles, i.e. the peak concentration at noon" to "and reproduces its diurnal cycle with the peak concentration occurring at noon"
Corrected, see lines 39-40 in the revised manuscript.

Lines 36-39: This sentence quotes a number which presumably is an average, but then notes differences during nighttime which is awkward. The sentence needs to be reworded.
We have reworded this sentence, see lines 40-43 in the revised manuscript.

Line 39: Change "the observation to "the observations"
Corrected, see line 44 in the revised manuscript.

Line 40: Change "lead" to "leads". Change "of SNA" to "in SNA"
Corrected, see line 45 in the revised manuscript.

Line 41: Change "the cloud water" to "cloud water content"
Corrected. Similar errors in the revised manuscript have been corrected.

Line 42: Change "bias of SNA simulation" to "simulated SNA bias".
Corrected, see lines 47-48 in the revised manuscript.

Line 43: Change "of cloud water can lead to model bias" to "in cloud water content can contribute to model biases"
Corrected, see lines 48-49 in the revised manuscript.

Lines 43-44: This result is not very surprising but a useful exercise since most air

quality simulations tend to focus on uncertainties in emissions, trace gas and aerosol interstitial chemical mechanisms, and boundary layer depth. The effects of clouds are examined less frequently from a process-level perspective, it would be more pleasing to simulate the amount and spatial distribution of cloud water more accurately. That could be achieved in part by constraining the ambient meteorology using data assimilation rather than constraining cloud water content alone.

We agree, we have constrained the ambient meteorology in all experiments by using the embedded objective analysis programs (OBSGRID). The OBSGRID improves the model performance on simulating meteorological fields by incorporating information from observations, including pressure, air temperature, dew point temperature, wind direction and speed from the surface and radiosonde reports, as well as remote sensing techniques. NCEP ADP Global Surface Observational Weather Data (https://rda.ucar.edu/datasets/ds461.0/) are chosen as input observation data in OBSGRID to nudge the initial and boundary meteorological conditions and provide surface fields for surface-analysis-nudging FDDA. Therefore, the model bias of ambient meteorology has been reduced to some extent. However, the simulated cloud water content is still much lower than MODIS observations. We have added descriptions in the revised manuscript, see lines 181-187.

The atmospheric reanalysis data, such as the National Center for Environmental Prediction's (NCEP) Final Analysis (FNL) dataset (NCEP/FNL), the ECMWF's Fifth generation Reanalysis (ERA5), the Japanese 55-year Reanalysis (JRA-55), the Modern-Era Retrospective Analysis for Research and Applications Version 2 (MERRA-2), and the China Meteorological Administration Reanalysis data (CRA), can provide meteorological initial and boundary conditions for numerical model. As a synthetic product of the realistic atmosphere, the quality of atmospheric reanalysis data depends on the quality and amount of observations assimilated and the performance of forecast and assimilation models. Because of the limitation of assimilation systems and the inconsistency between the numerical model dynamic field and the thermodynamic field, cloud-related satellite observations are difficult to be assimilated into reanalysis data (White et al., 2017). Cloud properties and processes are mostly parameterized in the model, there are thus uncertainties in cloud fields in the atmospheric reanalysis data (Yao et al., 2020). Kuma et al. (2020) found that MERRA-2 underestimated low cloud and fog occurrence relative to the ship observations on average 18% in the Southern Ocean during summer. Yao et al. (2020) found that the ERA5 and CRA both underestimate global monthly mean cloud cover by ~10% and ~20%. Cloud properties biases in atmospheric reanalysis data can also result in uncertainties in the simulation of meteorology and chemistry. Therefore, data assimilation and directly constraining cloud water content with satellite observations should be used at the same time to improve the cloud fields prediction in the future.

Lines 113-114: Change "the size distribution" to "droplet size distribution"
Corrected, see line 138 in the revised manuscript.

Line 133: Change "Model" to "model".
Corrected, see line 159 in the revised manuscript.

Line 141: Change "with the horizontal resolution" to "with a grid spacing". Grid spacing and resolution are different things. A model can resolve phenomena that are bigger than ~5 to 6 times the grid spacing.
Corrected, see lines 169-170 in the revised manuscript.

Line 142: Change "with 9 km" to "with a grid spacing of 9 km"
Corrected, see lines 169-170 in the revised manuscript.

Line 144: Change "parameterization schemes" to "parameterizations". The two words are redundant. The rest of the sentence switches between "parameterization" or "scheme" for unknown reasons. Pick one and be consistent.
Corrected, see lines 173 and 177 in the revised manuscript.

Line 151: Change "and Model" to "and the Model"
Corrected, see line 189 in the revised manuscript.

Line 159: Change "condition" to "conditions".
Corrected, see line 197 in the revised manuscript.

Line 162: change "as the" to "used as a"
Corrected, see line 201 in the revised manuscript.

Lines 171-177: Detail chemistry measurements are only available at two sites, so issues of spatial representativeness need to be address when evaluating the performance of the model. I do not doubt that cloud water is an important factor, but uncertainties in ambient conditions could displace plumes and contribute to model error. The model compares well to surface observations, but there is not assessment of how the model performs aloft which is important as well.
Thanks for suggestions. To evaluate the model performance on simulating surface air pollutants for a large region, we compared the simulated air pollutants ($SO_2$, $NO_2$, $PM_{2.5}$) concentrations with ground observations from the China National Environmental Monitoring Center in 8 typical cities, including Nanjing, Shanghai, Hangzhou, Hefei, Xuzhou, Heze, Linyi, and Lianyungang, which all experienced the haze-fog event (Fig. 2). We found that the model can reproduce the magnitude of observed daily air pollutants concentrations, and the correlation coefficients for $SO_2$, $NO_2$, and $PM_{2.5}$ are 0.3, 0.7, and 0.5, respectively. The model overestimates both $SO_2$ and $PM_{2.5}$ concentrations by about 80%, while the simulated $NO_2$ bias is much lower with the NMB of 7%.

Since the observed vertical profiles of air pollutants are not available, we employed the tropospheric $NO_2$ vertical column density (VCD) from TROPOMI (TROPOspheric

Monitoring Instrument) to evaluate the model performance (Fig. S4). The TROPOMI instrument, aboard the European Space Agency (ESA) Sentinel-5 Precursor (S-5P) satellite, was launched on 13 October 2017. It provides almost daily global coverage of tropospheric column densities (denoted as columns) of $NO_2$ with an unprecedented horizontal spatial resolution of $3.5 \times 7$ km$^2$, and overpasses at about 13:30 local time (LT) (van Geffen et al., 2020). The level-2 daily gridded TROPOMI $NO_2$ data with quality controls including cloud-screened (cloud fraction below 30%) and quality-assured (qa_value above 0.50) (Bauwens et al., 2020) are used in this comparison. The averaging kernels (AK, defined as the altitude-dependent air mass factor) used in the retrieval algorithms are applied it in the inter-comparison between TROPOMI with simulated tropospheric $NO_2$ columns. The TROPOMI $NO_2$ are re-gridded to the model grid ($9 \times 9$ km$^2$) for comparison. It is seen that the model can reproduce the spatial distribution of observed $NO_2$ VCD but overestimate the observed $NO_2$ VCD by 27% in the simulation domain (Fig. S4).

We have added the above discussions and two figures (Fig. 2, Fig. S4) in the revised version. See lines 216-235, 286-296.

Lines 183-185: State the spatial grid spacing of the observations.
Thanks for the suggestion. See lines 243-245 in the revised manuscript.

Lines 186-196: While it is encouraging the simulated meteorology is reasonable for one surface site, the authors do not provide the context of what this means in terms of the larger-scale meteorology that is contributing to fog and clouds over a much wider area in which aqueous chemistry is occurring.
We agree with the reviewer. We compared the simulated meteorology with the observations in 8 typical cities in the YRD, including Nanjing, Hangzhou, Shanghai, Lianyungang, Linyi, Heze, Xuzhou, and Hefei, which all experienced the haze-fog event. It is seen that the model can reproduce the temporal variation of observed meteorological variables in all cities, such as T2, RH, WS10, and WD10, with correlation coefficients all larger than 0.85, 0.68, 0.45, and 0.40. The mean biases (MBs) and root-mean-square errors (RMSEs) of hourly T2, RH2, WS10, and WD10 are also small, with the absolute MBs all lower than 1.0 °C, 6.0%, 0.8 m s$^{-1}$, 36.1° (except WD10 in Shanghai), and RMSEs all lower than 2.2 °C, 10.8%, 1.1 m s$^{-1}$, 110.1° (Table 1). There was almost no precipitation during this period. Similarly, the simulated precipitation is also quite limited except on 2 December. Overall, the simulated meteorological fields are reasonable in the YRD.

We have added the above discussions and Fig. 1 in the revised version. See lines 203-206, 252-269.

Lines 256-257: This is the first place the authors discuss simulated pH. It is not clear in text how they arrived at pH since it is not a standard output of WRF-Chem. In Fig. S2 they state they use an offline ISORROPIA model to compute pH. But that is a different

thermodynamic module than the one in MOSAIC, so saying these values are the simulated pH is not correct. So all the assumptions in modeled pH they describe in the manuscript are questionable.

We use the same thermodynamic model, i.e., ISORROPIA II (Fountoukis and Nenes, 2007), to calculate the simulated and observed $PM_{2.5}$ pH. The calculation of $PM_{2.5}$ pH is dependent on the concentrations of aerosol components (i.e., $Na^+, SO_4^{2-}, NH_3^+, NO_3^-, Cl^-, Ca^{2+}, K^+, Mg^{2+}$) and meteorological variables (i.e., RH and temperature). As shown in Fig. S6, the model underestimates $PM_{2.5}$ pH by 0.8, which leads to the discrepancies of $TNH_4$ gas-particle partitioning.

We have added the above explanations in the revised version, see lines 365-370.

Lines 271-275: The authors describe fog in the text, but the figure is showing LWP which is a vertically integrated quantity. Thus, LWP may not reflect cloud water at the surface. The text implies all the LWP in Figure 5 is fog, but it is possible that parts of those regions are just clouds. If the authors which to describe the areas in Figure 5 as fog, then some additional explanation is needed or different quantities should be plotted.

We agree with the reviewer that the integral of entire layers of LWC in the model, namely LWP, cannot represent the simulated fog area. Therefore, we re-define the fog area, i.e., any model grids with LWC larger than 0.01 g kg$^{-1}$ are defined as fog pixels (Zhou and Du, 2010), and below 1500 m are integrated to calculate the simulated LWP, and LWP larger than 2 g m$^{-2}$ is identified as the fog area (Jia et al., 2019).

We have added the above explanations in lines 385-389 and re-plotted Fig. 5 in the revised version.

Lines 276-286: Now the authors focus on surface LWC which is more appropriate in terms of fog. The note that visibility is overestimated by the model, especially on the 27th. But they fail to note that the model seems to underestimate visibility on other days and in other regions, which would argue against their main hypothesis. The amount of sulfate resulting from aqueous chemistry is likely a multiday process so that what is observed at Nanjing is likely an accumulation of sulfate formed over the region that is advected over Nanjing. Another problem with this comparison is that visibility could just be due to high levels of pollution that may be enhanced by aerosol water – and not fog.

Although the visibility is underestimated in some areas, the statistics of average visibility over the region where the fog occurred shows that the model generally overestimates VIS during the haze-fog event, except on 26 and 29 November. The mean value of observed and simulated visibility is shown in the upper right corner of each panel in Fig. 6. We have re-plotted Fig. 6 and added the above discussions in the revised version, see lines 400-404.

Fig. S3 shows the spatial distribution of observed daily $SO_2$, $NO_2$, and $PM_{2.5}$ concentrations, as well as the simulated wind speed and direction at 10m from 26

November to 2 December in the YRD. The simulated wind speed at 10m (WS10) is lower than 2 m s$^{-1}$ in the YRD, and such a low wind speed may not conducive to the advection and diffusion of air pollutants. The easterly wind brings humid air over the ocean, which favors the aqueous-phase formation of sulfate in Nanjing. Therefore, combining the variation of observed air pollutants concentrations and simulated wind fields, the formation of sulfate in Nanjing is mainly attributed to local chemical reactions. We have added Fig. S3 and above discussions in the revised version, see lines 277-284.

According to the function of visibility, $\text{VIS[m]} = -1000\ln(0.02)/(144.7\text{LWC[g m}^{-3}])^{0.88}$ (Kunkel et al., 1984). If the aerosol pollution reaches the level that visibility less than 1000 m, the aerosol water content is required to be larger than 0.016 g m$^{-3}$. However, previous studies reported that the aerosol water content during winter rarely exceeded $10^{-3}$ g m$^{-3}$ (Wu et al., 2018; Shen et al., 2019). Therefore, the visibility less than 1000 m should be attributed to the fog.

Line 298: Differences in the spatial resolution makes it difficult to really assess the potential errors in the model. It would make more sense to average the simulated LWP over the same area as the MODIS grid cells for a more fair comparison. This would affect the fittings performed in the next section.
Thanks for suggestions. We averaged the simulated LWP to the MODIS grid cells, and then compared the simulated LWP with the MODIS observations. We have added the details in the revised version, see lines 248-250.

Line 318: I understand the rationale for modifying the cloud water only in the aqueous chemistry module. But the authors need to put this into context with their findings later in the paper. In reality, changing water content will ultimately impact other aspects of the coupled processes in the model and ultimately change predictions of aerosol species and aqueous chemistry.
We agree with the reviewer that changing cloud water content can affect other aspects of coupled processes in the model (i.e., radiative transfer, wet/dry deposition, photolysis rates, and gas-phase chemistry). Therefore, we conducted a sensitivity experiment, i.e., constraining the cloud water content in all above processes in the model (Sen_all_c_pH). Fig. S9 shows the difference of simulated SNA between the two simulations (Sen_c_pH and Sen_all_c_pH) during the haze-fog event in the YRD. Constraining the cloud water in all coupled processes causes the simulated sulfate, nitrate, and ammonium concentrations to decrease by 2.0, 1.8, and 7.4 μg m$^{-3}$ compared to the Sen_c_pH over the entire YRD, with the biggest difference all in the Jiangsu and Anhui provinces, indicating that the increase of cloud water content can increase the aerosol deposition flux and decrease the photolysis rate, and ultimately change predictions of aerosol species and aqueous-phase chemistry and oxidants concentrations.

We have added the above discussions and Fig. S9 in the revised version, see lines 463-

468 and 529-538.

Line 397-398: The authors state they investigated the possible reasons contributing to model bias, when in fact they only investigated two: cloud water and specification of pH. The authors need to be more clear and explicit here. The authors could have strengthened their argument by performing other sensitivity simulations that explore other uncertainties such as emissions and chemical mechanisms and putting those results into the context of the cloud water sensitivity analysis.

Thanks for suggestions, we have revised in the revised manuscript, see lines 553-555.

In our previous work, we studied the sulfate underestimation caused by the uncertainties of emissions and chemical mechanisms, including gas-phase chemistry and heterogeneous reaction in the model.

Sha et al. (2019a) suggested that different emission inventories ("bottom-up" and "top-down") result in a 4 $\mu$g m$^{-3}$ (60%) difference in the simulated sulfate concentrations during a haze event in Shanghai, while the simulated sulfate using both inventories are much lower than the observations. Therefore, the incomplete and/or inaccurate chemical mechanism in the model might be another main reason for the underestimation of sulfate. Generally, sulfate is formed through the gas-phase oxidation of SO$_2$ by OH radicals, and aqueous-phase oxidation of S(IV) ( = SO$_2$·H$_2$O+HSO$_3^-$+SO$_3^{2-}$) by various oxidants (e.g., H$_2$O$_2$, O$_3$, NO$_2$, and O$_2$ (transition-metal-ion (TMI) catalysis)) in cloud droplets and aerosol water (the reactions in aerosol water often called heterogeneous reaction) (Cheng et al., 2016; Liu et al., 2020). By conducting several sensitivity experiments, our recent work showed that tripling the gas-phase oxidation rate of SO$_2$ by OH only enhances sulfate by 72% during winter in Nanjing, still 73% lower than the observations, indicating gas-phase oxidation is possibly not the major causes for the underestimations in the model (Sha et al., 2019b).

To tackle the underestimation of sulfate in the model during haze events, some studies added SO$_2$ heterogeneous reactions in the model usually parameterizing as a reactive uptake process and assuming to be irreversible (Wang et al., 2014; Zheng et al., 2015; Chen et al., 2016; Li et al., 2017; Feng et al., 2018; Li et al., 2018; Shao et al., 2019). Although the implementation of SO$_2$ heterogeneous reactions in the model could achieve an agreement of simulated and observed sulfate concentrations, the model still underestimates sulfate. Our recent study showed that the simulated sulfate was 53% lower than the observations during a haze event in Nanjing when including SO$_2$ heterogeneous reaction in aerosol water (Sha et al., 2019b). This is mainly due to uncertainties of the parameters in this reaction, such as the pH, water content, the surface area of aerosol, and the gas uptake coefficients on aerosol water.

According to our previous work, we have added the results and findings that the contribution of other uncertainties such as emissions and chemical mechanisms to the model bias, and put these in the Introduction in the revised manuscript, see lines 82-87,

Lines 431-434: It would have been useful to provide some more reflection of what could be done to improve the cloud predictions. For example, data assimilation is often used to constrain the ambient meteorology which often leads to improved cloud predictions. That is a tool that can be used more routinely for air quality applications and is more pleasing than brute force adjusting cloud water content. Exploring other microphysics representation is another option.

Thanks for suggestions, we have added more reflections of what could be done to improve the cloud predictions in the Conclusion in the revised manuscript, see lines 597-602.

**Reference:**

Bauwens, M., Compernolle, S., Stavrakou, T., Muller, J. F., van Gent, J., Eskes, H.; Levelt, P. F., van der, A. R., Veefkind, J. P., Vlietinck, J., Yu, H., and Zehner, C. Impact of coronavirus outbreak on NO2 pollution assessed using TROPOMI and OMI observations, Geophys. Res. Lett., 47, e2020GL087978, 2020.

Chen, D., Liu, Z. Q., Fast, J., and Ban, J. M.: Simulations of sulfate–nitrate–ammonium (SNA) aerosols during the extreme haze events over northern China in October 2014, Atmos. Chem. Phys., 16 (16), 10707-10724, 2016.

Cheng, Y. F., Zheng, G. J., Wei, C., Mu, Q., Zheng, B., Wang, Z. B., Gao, M., Zhang, Q., He, K. B., Gregory, R. C., Ulrich, P., and Su, H.: Reactive nitrogen chemistry in aerosol water as a source of sulfate during haze events in China, Sci. Adv. 2 (12) 1601530- 1601530, 2016.

Feng, T., Bei, N. F., Zhao, S. Y., Wu, J. R., Li, X., Zhang, T., Cao, J. J., Zhou, W. J., and Li, G. H.: Wintertime nitrate formation during haze days in the Guanzhong basin, China: a case study, Environ. Pollut. 243, 1057–1067, 2018.

Fountoukis, C. and Nenes, A.: ISORROPIA II: a computationally efficient thermodynamic equilibrium model for $K^+ - Ca^{2+} - Mg^{2+} - NH_4^+ - Na^+ - SO_4^{2-} - NO_3^- - Cl^- - H_2O$ aerosols, Atmos. Chem. Phys. 7, 4639–4659, 2007.

Jia, X. C., Quan, J. N., Zheng, Z. Y., Liu, X. G., Liu, Q., He, H., and Liu, Y. G.: Impacts of anthropogenic aerosols on fog in North China Plain, J. Geophys. Res. Atmos., 124, 252–265. https://doi.org/10.1029/2018JD029437, 2019.

Kuma, P., McDonald, A. J., Morgenstern, O., Alexander, S. P., Cassano, J. J., Garrett, S., Halla, J., Hartery, S., Harvey, M. J., Parsons, S., Plank, G., Varma, V., and Williams, J.: Evaluation of Southern Ocean cloud in the HadGEM3 general circulation model and MERRA-2 reanalysis using ship-based observations, Atmos. Chem. Phys., 20, 6607–6630, https://doi.org/10.5194/acp-20-6607-2020, 2020.

Kunkel, B.A.: Parameterization of droplet terminal velocity and extinction coefficient in fog models, J. Clim. Appl. Meteorol. 23, 34–41, https://doi.org/10.1175/1520-0450(1984)023<0034:PODTVA>2.0.CO;2, 1984.

Li, G. H., Bei, N. F., Cao, J. J., Hu, R. J., Wang, J. R., Feng, T., Wang, Y. C., Liu, S. X., Zhang, Q., Tie, X. X., and Molina, L. T.: A possible pathway for rapid growth of sulfate during haze days in China, Atmos. Chem. Phys., 17, 3301–3316, 2017.

Li, J., Chen, X. S., Wang, Z. F., Du, H. Y., Yang, W. Y., Sun, Y. L., Hu, B., Li, J. J., Wang, W., Wang, T., Fu, P. Q., and Huang, H. L.: Radiative and heterogeneous chemical effects of aerosols on ozone and inorganic aerosols over East Asia, Sci. Total Environ., 622–623, 1327–1342, 2018.

Liu, P. F., Ye, C., Xue, C. Y., Zhang, C. L., Mu, Y. J., and Sun, X.: Formation mechanisms of atmospheric nitrate and sulfate during the winter haze pollution periods in Beijing: gas-phase, heterogeneous and aqueous-phase chemistry, Atmos. Chem. Phys., 20, 4153–4165, https://doi.org/10.5194/acp-20-4153-2020, 2020.

Sha T., Ma X. Y., Jia H. L., Tian R., Chang Y. H., Cao F., and Zhang Y. L.: Aerosol chemical component: Simulations with WRF-Chem and comparison with observations in Nanjing, Atmos. Environ., 218, 116982,

https://doi.org/10.1016/j.atmosenv.2019.116982, 2019b.

Sha, T., Ma, X. Y., Jia, H. L., van der A, R. J., Ding, J. Y., Zhang, Y. L., and Chang, Y. H.: Exploring the influence of two inventories on simulated air pollutants during winter over the Yangtze River Delta, Atmos. Environ., 206, 170–182, 2019a.

Shao, J. Y., Chen, Q. J., Wang, Y. X., Lu, X., He, P. Z., Sun, Y. L., Shah, V., Martin, R. V., Philip, S., Song, S. J., Zhao, Y., Xie, Z. Q., Zhang, L., and Alexander, B.: Heterogeneous sulfate aerosol formation mechanisms during wintertime Chinese haze events: air quality model assessment using observations of sulfate oxygen isotopes in Beijing, Atmos. Chem. Phys., 19, 6107–6123, 2019.

Shen, X. J., Sun, J. Y., Zhang, X. Y., Zhang, Y. M., Zhong, J. T., Wang, X., Wang, Y. Q., and Xia, C.: Variations in submicron aerosol liquid water content and the contribution of chemical components during heavy aerosol pollution episodes in winter in Beijing, Sci. Total Environ., 693, 133521, DOI: 10.1016/j.scitotenv.2019.07.327, 2019.

van Geffen, J., Boersma, K. F., Eskes, H., Sneep, M., ter Linden, M., Zara, M., and Veefkind, J. P.: S5P TROPOMI NO2 slant column retrieval: method, stability, uncertainties and comparisons with OMI., Atmos. Meas. Tech., 13, (3), 1315-1335, 2020.

Wang, Y. X., Zhang, Q. Q., Jiang, J. K., Zhou, W., Wang, B. Y., He, K. B., Duan, F. K., Zhang, Q., Philip, S., and Xie, Y. Y.: Enhanced sulfate formation during China's severe winter haze episode in January 2013 missing from current models, J. Geophys. Res. Atmos., 770 119, 10425-10440, https://doi.org/10.1002/2013JD021426, 2014.

White, A. T., Pour-Biazar, A., Doty, K., Dornblaser, B., and McNider, R. T.: Improving cloud simulation for air quality studies through assimilation of Geostationary Satellite Observations in retrospective meteorological modeling, Mon. Weather Rev., 146(1), 29–48, https://doi.org/10.1175/MWR-D-17-0139.1, 2017.

Wu, Z., Wang, Y., Tan, T., Zhu, Y., Li, M., Shang, D., Wang, H., Lu, K., Guo, S., Zeng, L., and Zhang, Y.: Aerosol liquid water driven by anthropogenic inorganic salts: implying its key role in haze formation over the North China Plain, Environ. Sci. Tech. Lett., 5, 160–166, https://doi.org/10.1021/acs.estlett.8b00021, 2018.

Yao, B., Teng, S.W., Lai, R.Z., Xu, X.F., Yin, Y., Shi, C.X., and Liu, C.: Can atmospheric reanalyses (CRA and ERA5) represent cloud spatiotemporal characteristics?, Atmos. Res., 244, 105091, 2020.

Zheng, G. J., Duan, F. K., Su, H., Ma, Y. L., Cheng, Y., Zheng, B., Zhang, Q., Huang, T., Kimoto, T., Chang, D., Poschl, U., Cheng, Y. F., and He, K. B.: Exploring the severe winter haze in Beijing: the impact of synoptic weather, regional transport and heterogeneous reactions, Atmos. Chem. Phys., 15, 2969-2983, https://doi.org/10.5194/acp-15-2969-2015, 2015.

Zhou, B. and Du, J.: Fog prediction from a multimodel mesoscale ensemble prediction system, Weather Forecast., 25(1), 303–322. https://doi.org/10.1175/2009WAF2222289.1, 2010.

---

## Author Comment (AC2) · 3 Jan 2021

Thanks to the referee for his/her very thoughtful suggestions. Below we address the reviewers' comments, with the reviewer comments in black, and our response in blue. We have revised the manuscript accordingly, and mentioned the line number of the **tracked revision**.

**Anonymous Referee #2:**

**General comments:**

East Asian countries and regions are always suffered from serious air pollutions with rapid economic growth in recent decades. And high level emissions of air pollutants in East Asia could further affect regional air qualities, human health, traffic safeties as well as regional or global climate changes. Observations have revealed that severe and persistent haze pollutions occurred frequently in China during recent years. Although the numerical models could capture the loading levels and temporal-spatial variations of the total PM, most of them could not well simulate their chemical components, especially in heavy pollution episodes. Thus, accurately predicting the concentrations and chemical components of particulate matter are still very challenging for climate and air quality models. In this study, influence of aqueous-phase chemistry on the formation of near surface sulfate as well as the concentrations of total ammonium is carried out to investigate the importance of this process in some polluted episodes, based on observations and numerical evolutions. Therefore, the topic of this study is interesting and novel to some degrees and the paper has a potential for publication in the journal.

Thanks to the reviewer for the comments and suggestions.

**Specific Comments:**

1. Both Abstract and Conclusions should be more concise, instead of only repeating the results.

We have revised the Abstract and Conclusions.

2. Were the aerosol or trace gases from biomass burning taken into account in the simulations? What is the resolution of the emission inventory (MEIC)? Why the emissions in 2016 were used to assess the pollution episode in 2018?

According to previous studies (Du et al., 2017; Wu et al., 2018), the peak emissions of biomass burning are normally found in summer and autumn harvest periods, including May, June, September, and October, while the contribution of biomass burning emission to the pollutants in winter is generally low. From the MODIS Fire and Thermal Anomalies product, the detective hotspots are sparse during this haze-fog event in the YRD, thus the contribution of biomass burning emission is deemed to be minimal in this study case (Fig. R1). Therefore, the aerosols and trace gases from biomass burning are not considered in the simulations.

To quantify the impact of biomass burning emission on aerosols and trace gases concentrations, we conducted an additional simulation with biomass burning emission

(i.e., BBE). The biomass burning emission is taken from Fire INventory from NCAR (FINN), including the emissions of CO2, CO, NO, NO2, SO2, NH3, OC, BC, PM, CH4, and NMVOC, etc. From Fig. R2, the emissions of SO2, NOx, and NH3 from biomass burning are distributed sporadically in the YRD, with the total emissions less than 219 kg km-2 for SO2, 254 N kg km-2 for NOx, 491 kg km-2 for NH3 during this period. The contribution of biomass burning to total SO2, NO2, and NH3 emissions is zero in most areas. Only in some small places, biomass burning accounts for half of the total emissions. Additionally, we quantify the impacts of biomass burning emission on the simulated SO2, NO2, NH3, and PM2.5 (Fig. R3). Biomass burning emissions only change the mean concentrations of air pollutants (SO2, NO2, NH3, SNA, and PM2.5) by less than 10% during this period in the YRD. Therefore, biomass burning emissions have little impact on air pollutants concentrations during this period, and we do not need to consider biomass burning emissions in this study.

Figure R1. MODIS hotspots from 26 November to 2 December in the YRD.

---

## Author Comment (AC3) · 3 Jan 2021

Thanks to the referee for his/her very thoughtful suggestions. Below we address the reviewers' comments, with the reviewer comments in black, and our response in blue. We have revised the manuscript accordingly, and mentioned the line number of the **tracked revision**.

**Anonymous Referee #3:**

**General comments:**

The chemical transport model is an important tool for the study of air pollution and emission control. The ability of the model to simulate aerosol and its components is an important standard to evaluate the model This manuscript evaluated the WRF-Chem performance on simulating inorganic aerosol components of PM2.5 during a haze-fog event in Nanjing, and investigate the possible reasons of simulating bias compared with the observations. It found that the strong sensitivity of SNA concentration to the cloud water provides an explanation for the bias of SNA simulation. The topic is of interest and the manuscript is generally well written. There are several issues that need to be addressed before the manuscript can be accepted for publication.

Thanks to the reviewer for the comments and suggestions.

**Specific Comments:**

1. Because LWP is a vertically integrated quantity. Is the large of MODIS LWP possible due to the thickness of the fog is thicker? I think the effect of the vertical profile simulation can be compared. If there is no observation data, vertical sounding and simulation can be compared.

Thanks for suggestions, we agree with the reviewer that the underestimation of LWP may also be caused by the simulated fog thickness being thinner than the observed.

Since we have not found the vertical sounding of cloud water content during this period in Nanjing, so we compared the vertical profile of simulated relative humidity (RH) in the Control run with the observations from Zou et al. (2020) during this haze-fog event (Fig. R1). The observed RH was collected at the Station for Observing Regional Process of the Earth System (SORPES). Due to the lack of other vertical observational data, we set the thickness of fog roughly to a height where the RH is less than 90%. We found that the model underestimates the thickness of fog at 08:00 LT on 26 and 30 November by about 50 m and 150 m but the thickness of simulated fog at 08:00 LT is generally reasonable. Therefore, the deviation of simulated fog thickness is not the main reason that the simulated LWP is lower than the MODIS observations.